

# Sulfate deprivation triggers high methane production in a disturbed and rewetted coastal peatland

Franziska Koebsch[1,2], Matthias Winkel[1], Susanne Liebner[1,3], Bo Liu[4,5], Julia Westphal[4], Iris Schmiedinger[4], Alejandro Spitzy[6], Matthias Gehre[7], Gerald Jurasinski[2], Stefan Köhler[2], Viktoria Unger[2], Marian Koch[2,8], Torsten Sachs[1], Michael E. Böttcher[4]

[1]GFZ German Research Centre for Geosciences, 14473 Potsdam, Germany
[2]Department for Landscape Ecology and Site Evaluation, University of Rostock, 18059 Rostock, Germany
[3]Institute of Biochemistry and Biology, University of Potsdam, 14476 Golm, Germany
[4]Geochemistry and Isotope Biogeochemistry Group, Leibniz Institute for Baltic Sea Research (IOW), 18119 Warnemünde, Germany
[5]Section Marine Geochemistry, Alfred Wegener Institute Helmhotz Center for Polar and Marine Research, Am Handelshafen 12, 27570 Bremerhaven, Germany
[6]Institute for Geology, Biogeochemistry Department, University of Hamburg, 20146 Hamburg, Germany
[7]Department of Isotope Biogeochemistry, Helmholtz Centre for Environmental Research UFZ, 04318 Leipzig, Germany
[8]Tropical Plant Production and Agricultural Systems Modelling, University of Göttingen, 37073 Göttingen, Germany

*Correspondence to*: Franziska Koebsch (Franziska.koebsch@uni-rostock.de)

**Abstract.** In natural coastal wetlands, high supplies of marine sulfate suppress methanogenesis. Coastal wetlands are, however, often subject to disturbance by dyking and drainage for agricultural use and it has been shown that they can turn to potent methane sources when rewetted for remediation, suggesting that the sulfate-related methane suppressing mechanisms were suspended by the preceding land use measures. Here, we unravel the hydrological relocation and biogeochemical S and C transformation processes that induced high methane emissions in a disturbed and rewetted peatland despite former brackish impact. The underlying processes were investigated along a transect of increasing distance to the coastline using a combination of concentration patterns, stable isotope partitioning and analysis of the microbial community structure. We found that dyking and freshwater rewetting caused a distinct freshening and an efficient depletion of the brackish sulfate reservoir by dissimilatory sulfate reduction (DSR). Despite some legacy effects of brackish impact expressed as high amounts of sedimentary S and elevated electrical conductivities, contemporary metabolic processes operated mainly under sulfate-limited conditions. This opened up favorable conditions for the establishment of a prospering methanogenic community in the top 30-40 cm of peat, the structure and physiology of which resembles those of terrestrial organic-rich environments. Locally, high amounts of sulfate persisted in deeper peat layers through the suppression of DSR, probably by competitive electron acceptors of terrestrial origin, for example Fe(III), but did not interfere with high methane emissions on ecosystem scale. Our results indicate that the climate effect of disturbed and remediated coastal wetlands cannot simply be derived by analogy with their natural counterparts. From a greenhouse gas perspective, the re-exposure of dyked wetlands to natural coastal dynamics would literally open up the floodgates for a replenishment of the marine sulfate pool and constitute an efficient measure to reduce methane emissions.



## 1. Introduction

Coastal wetlands play an important role in climate change mitigation and adaption as they can efficiently accrete organic sediments, adjust coastal elevations to sea level rise and protect low-lying areas in the hinterland. Further, while freshwater wetlands constitute the largest natural source of the greenhouse gas methane (Zhang et al., 2017), the efficient accumulation of autochthonous C in coastal wetlands comes without the expense of high $CH_4$ emissions (Holm et al., 2016). Methane is a potent greenhouse gas that is formed as terminal product of organic matter breakdown under strictly anaerobic conditions typically in the absence of electron acceptors other than carbon dioxide ($CO_2$) (Segers and Kengen, 1998). In coastal environments, methane production and emission are effectively suppressed by sulfate-rich seawaters: methanogens are outcompeted by sulfate reducing bacteria (SRB) for acetate-type precursors and hydrogen (Lovley and Klug, 1983; Schönheit et al., 1982). This shifts the prevailing anaerobic C metabolic pathways from methanogenesis towards dissimilatory sulfate reduction of dissolved organic matter (DSR) (King and Wiebe, 1980; Martens and Berner, 1974). In addition, sulfate operates as electron acceptor for anaerobic methane oxidation by a syntrophic consortium of anaerobic methanotrophs (ANME) and SRB (Boetius et al., 2000; Iversen and Jorgensen, 1985). Anaerobic methane oxidation has been specifically described for brackish wetland sediments, but is not exclusively confined to the utilization of sulfate as electron acceptor (Segarra et al., 2015; Segarra et al., 2013).

Human activities such as dyking and drainage place intensive pressure on coastal landscapes with sometimes irreversible impairments of their biogeochemical cycles and ecosystem functions (Karstens et al., 2016; Zhao et al., 2016). Dykes separate coastal wetlands from resupply of seawater, and drainage for agricultural use induces the aerobic decomposition of organic-rich sediments, resulting in substantial $CO_2$ losses and land subsidence (Deverel and Rojstaczer, 1996; Miller, 2011). As sea levels are expected to rise, the controlled retreat from flood-prone areas becomes an essential strategy of integral coastal risk management to complement conventional technical solutions such as dyking (Sánchez-Arcilla et al., 2016). Rewetting may re-establish the ability of abandoned coastal wetlands to efficiently accrete organic matter under anaerobic conditions and represents a promising management technique to reverse land surface subsidence caused by drainage-induced peat oxidation (Deverel et al., 2016). Moreover, while freshwater wetlands may become methane sources upon rewetting (Franz et al., 2016; Hemes et al., 2018; Vanselow-Algan et al., 2015; Wilson et al., 2009), sulfate-rich seawater could potentially reduce post-rewetting methane release in coastal wetlands. However, recent work on a degraded brackish peatland has revealed high post-rewetting $CH_4$ emissions (Hahn et al., 2015; Wen et al., in review, 2018) and methanogen abundance {Wen & Unger, 2018) thereby challenging the common notion of coastal wetlands as negligible methane emitters. In fact, dyking and the drainage-rewetting cycle may induce hydrodynamic shifts and biogeochemical transformation processes that are so far not well understood. In particular, the transformation and/or relocation of the marine sulfate reservoir in the sediments of dyked wetlands are of vital importance to understand the implications of anthropogenic intervention on coastal wetland biogeochemistry and to better constrain the climate effect of coastal wetland remediation.



Here, we investigate the mechanisms that allow for high methane production in disturbed and remediated coastal wetlands. We therefore address the fate of brackish compounds and the emerging S and C transformation processes in a rewetted, freshwater-fed peatland that was naturally exposed to episodic intrusions from the Baltic Sea. In the past, the peatland had

been subject to intense human intervention including dyking and drainage for agricultural use. After rewetting by freshwater-flooding, the site turned into a strong methane source. The underlying hydrological and biogeochemical processes were investigated along a brackish-terrestrial transect that spans between 300 and 1,500 m distance from the coastline using hydrogeochemical element patterns, stable isotope biogeochemistry and microbiological analyses

The specific goals were to:

- retrace the marine legacy effect remaining after dyking and freshwater rewetting in the peat pore space using salinity, the isotope composition of water and a suite of dissolved constituents that may be indicative for the intermingling of brackish or terrestrial impact

- track the distribution of Baltic Sea-derived pore water sulfate in the peatland soil based on major geochemical gradients delineated by brackish and freshwater tracers

- uncover potential S transformation pathways using concentration and stable isotope measurements of pore water $SO_4^2$ ($\delta^{34}S$ and $\delta^{18}O$) and solid S compounds interpreted in the light of the bacterial community structure and the presence of sulfate-reducing bacteria

- describe dissimilatory C decomposition pathways and methane cycling processes in relation to the found S transformation patterns based on concentration and stable isotope measurements of $CH_4$ ($\delta^{13}C$, $\delta^2H$) and dissolved

inorganic C (DIC, $\delta^{13}C$) as well as to the abundance and community structure of methanogenic and methanotrophic archaea

We hypothesized the marine legacy effect to express as lateral gradient in electrical conductivity (EC) and pore water sulfate along the brackish-terrestrial transect. We further expected increasing terrestrial impact to promote the deprivation of the brackish sulfate pool and to induce complementary patterns of methane production.

## 2. Material and Methods

### 2.1 Study site and sampling design

The study site is part of the nature reserve 'Heiligensee und Hütelmoor', a 490 ha coastal peatland complex located in NE Germany directly at the SW Baltic coast with an elevation between -0.3 and + 0.7 m above sea level (Dahms, 1991) (latitude 54°12', longitude 12°10', Fig. 1). Climate is transitional maritime with continental influence from the east. The area receives

a mean annual precipitation of 645 mm with a mean annual temperature of 9.2°C (reference period 1982-2011, data from the German Weather Service (DWD)). Peat formation was initiated by the Littorina Sea transgression and the post-glacial sea level rise around 5400 BC. Presently, the Hütelmoor is fed by a 15 km² forested catchment dominated by gley over fine sands. Originally, the fen exhibited 0.2-2.3 m deep layers of sulfidic reed-sedge peat underlain by Late Weichselian sands over





impermeable till (Bohne and Bohne, 2008; Voigtländer et al., 1996). Forty years of drainage for grassland use caused severe

degradation of the peat, which was recently identified as sapric histosol (Koebsch et al., 2013). Since the rewetting by flooding in 2010 through the construction of a weir at the outflow of the catchment, more than 80% of the area have been permanently inundated with freshwater from the surrounding forest catchment (Miegel et al., 2016). Current vegetation of the Hütelmoor is dominated by patches of competitive emergent macrophytes such as reed and sedges (Common Reed (*Phragmites australis (Cav.) Trin. ex Steud* and *Carex acutiformis Ehrh.)* that increasingly supersede species indicative for brackish conditions

(*Bolboschoenus maritimus (L.)*, *Palla Schoenoplectus tabernaemontani (C. C. Gmel.) Palla*) (Koch et al., 2017).

Under natural coastal dynamics, the Hütelmoor is episodically flooded by storm surges. Low outflow and high evapotranspiration rates promote brackish conditions. Major brackish water intrusions were reported for 1904, 1913, 1949, 1954 and 1995 (Bohne and Bohne, 2008) though flooding frequency is reduced since the site was dyked in 1903. Additional brackish input occurs through underground flow and atmospheric deposition as well as through high water situations at the

Baltic Sea when backwater of the interconnected Warnow river delta enters the fen. However, potential brackish water entry paths other than storm surges have revealed negligible effect on peat salinity (Selle et al., 2016). The last flooding event in 1995 raised EC in the drainage ditches up to 8 mS cm$^{-1}$, but the EC decreased to the pre-flooding level of 2 mS cm$^{-1}$ within the following five years (Bohne and Bohne, 2008).

Samples were collected at four spots along a transect with increasing distance to the Baltic Sea (300-1,500m, Fig. 1b) within

two weeks in October/November 2014. The transect included the area of a former study which revealed high concentrations of brackish $SO_4^{2-}$ with annual means up to 23.7±3.2 mM (unpublished, Fig. 1c). At the time of sampling, water depth above peat surface spanned from 15 to 25 cm, which presented the lowest range within the seasonal water level fluctuation. Sampling depth ranged from 45 to 65 cm which was in most cases sufficient to cover the full peat depth incl. the underlying mineral soil.

### 2.2    Pore water analysis

Pore waters were collected from distinct depth below the surface (cmbsf.) with a stainless steel push-point sampler attached to a syringe to draw the sample from a distinct penetration depth. Temperature, EC and salinity were measured directly after sampling (Sentix 41 pH probe and a TetraCon 325 conductivity measuring cell attached to a WTW multi 340i handheld; WTW, Weilheim). Samples were filtered (0.45 µm membrane syringe filters) in situ and transferred without headspace into vials (except for dissolved $CH_4$). Vials had been previously preconditioned with 1 M HCl and subsequent 1 M NaOH and were

filled with a compound-specific preservative (see below).

Dissolved $CH_4$ concentration was measured with the headspace approach. Therefore, 5 ml of pore water were transferred into 12 ml septum-capped glass vials under atmospheric pressure. Before taking them to the field, the sampling vials were flushed with Ar and filled with 500 µl saturated $HgCl$ solution to prevent further biological activity. After sampling, the punctuated septum was covered with lab foil and the vials were stored upside down to minimize $CH_4$ loss. Headspace gas concentrations

after equilibration were measured in duplicates with an Agilent 7890A gas chromatograph equipped with a flame ionization detector and with a carbon plot capillary column or HP-Plot Q (Porapak-Q) column. Helium was used as tracer gas. Gas sample




analyses were performed after calibration of the gas chromatograph with standard gas that achieved reproducibility > 98.5%. The measured headspace $CH_4$ concentration was then converted into dissolved $CH_4$ concentration using the temperature-corrected solubility coefficient (Wilhelm et al., 1977).

Samples for anion concentrations ($SO_4^{2-}$, $Cl^-$, $Br^-$) were filled in 20 ml glas vials preserved with 1 ml 5% ZnAc-solution to prevent sulfide oxidation. Anion concentrations were analyzed by IC (Thermo Scientific Dionex) in a continuous flow of 9 mM $NaCO_3$ eluent in an Ion Pac AS-9-HC 4 column, partly after dilution of the sample. The device was calibrated with NIST SRM standard solutions freshly prepared before each run to span the concentration ranges of the (diluted) samples. Reproducibility between sample replicates was better than ±5%.

For $H_2S$ analysis, pore water was filled into 5 ml polypropylene vials and preserved with 0.25 ml 5% ZnAc solution. $H_2S$ concentration was measured photometrically (Specord 40, Analytic Jena) using the methylene blue method (Cline, 1969). The metal and total S concentrations were analysed by ICP-OES (iCAP 6300 DUO Thermo Fisher Scientific) after appropriate dilution. Since high amounts of DOC may cause severe interferences in the ICP-OES element measurements, samples were boiled in Teflon beakers with 65% $HNO_3$ and subsequent 19% HCl prior to analysis. The accuracy and precision was routinely

checked with the certified CASS standards as described previously (Kowalski et al., 2012).

$\delta^{13}C$ and $\delta D$ values of methane were analyzed using the gas chromatography-combustion-technique (GC-C) and the gas chromatography-high-temperature-conversion-technique (GC-HTC). The gas was direct injected in a Gas Chromatograph Agilent 7890 (Agilent Technologies, Germany), the peaks were separated using a CP-PoraBOND Q GC-column (50mx0.32mmx5µm, isotherm 60°C, Varian). Methane was quantitatively converted to the analysis gases $CO_2$ and $H_2$ in the

GC-Isolink-Interface (Thermo Finnigan, Germany) and directly transferred via open split interface (ConFlo IV, Thermo Finnigan, Germany). The $\delta^{13}C$ and $\delta D$ values of both gases were then measured with the isotope-ratio-mass-spectrometer MAT-253 (Thermo Finnigan, Germany). Results for $\delta^{13}C$ ratios of methane are given in the usual δ-notation versus the Vienna PeeDee Belemnite (VPDB) standard. $\delta D\text{-}CH_4$ ratios were referenced to the Vienna Standard Mean Ocean Water (V-SMOW). The carbon isotope values ($\delta^{13}C$) of DIC were measured from a HgCl-preserved solution using a Thermo Finnigan MAT 253

gas mass spectrometer coupled to a Thermo Electron Gas Bench II via a Thermo Electron Conflo IV split interface. NBS19 and LSVEC were used to scale the isotope measurements to the VPDB standard. Based on replicate measurements of standards, reproducibility was better than ±0.1‰ (Winde et al., 2014). The stable carbon isotope ratio of DOC ($\delta^{13}C$) was determined according to Ertl and Spitzy (2004), involving cryogenic trapping of $CO_2$ and isotope ratio mass spectrometry with a Finnigan Mat 252 with dual-inlet system. A modified combustion module was coupled on-line to the cryogenic trap: before trapping, a

20 ml sample was combusted by way of continuous injection (0.85 ml min$^{-1}$) in a helium stream into a self-assembled high temperature catalytic oxidation unit, consisting of a furnace heated to 950 ºC and a quartz glass column filled with copper oxide and cerium oxide. Combustion gases were dried using Peltier coolers and a magnesia perchlorate trap. $^{13}C/^{12}C$ values (‰) were obtained from at least duplicate analyses and referenced to the VPDB standard. The standard deviation was smaller than 0.5‰.




For the determination of sulfate isotope signatures, dissolved sulfate was precipitated with 5% barium chloride as barium
sulfate as described in Böttcher et al. (2007). After precipitation the solid was filtered, washed and dried, and further combusted
in a Thermo Flash 2000 EA elemental analyzer that was connected to a Thermo Finnigan MAT 253 gas mass spectrometer via
a Thermo Electron Conflo IV split interface with a precision of better than ±0.2 ‰. Isotope ratios are converted to the VCDT
scale following Mann et al. (2009). For oxygen isotope analyses, $BaSO_4$ was decomposed by means of pyrolysis in silver cups
using a high temperature conversion Elemental Analyzer (HTO-, Hekatech, Germany) connected to an isotope gas mass
spectrometer (Thermo FinniganMAT 253) at the Helmholtz Centre for Environmental Research – UFZ according to the
method described by Kornexl et al. (1999). The calibration took place via the reference materials IAEA-SO-5 and -SO-6 and
$^{18}O/^{16}O$ values were referenced to the V-SMOW standard. Replicate measurements agreed within ± 0.5‰.

Stable oxygen (O) pore water isotope measurements were conducted using a CRDS system (Picarro L2140-i) versus the V-
SMOW standard. International V-SMOW, SLAP, and GISP, besides in-house standards were used to scale the isotope
measurements. The δ-values are equivalent to milli Urey (mU; Brand and Coplen 2012).

### 2.3  Sediment analysis

Intact peat cores were collected with a perspex liner (ID: 59.5 mm) and subsequently punched out layer-by-layer. The peat
section protruding from the end of the liner was divided into 3 subsamples for the analysis of (i) Total reduced inorganic S
(TRIS), (ii) total S (TS) and reactive iron, and (iii) the microbial community structure. In order to minimize oxygen
contamination, the outer layer of the peat core was omitted and subsamples were immediately packed. The aliquot for TRIS
analysis was preserved with 1:1 (v/v) 20% ZnAc. Subsamples for microbial analysis were immediately stored in RNAlater to
preserve DNA and RNA. A second core was taken for the analysis of water content and dry bulk density. TS and TRIS samples
were frozen within 8 hrs after collection. Aliquots for TS elemental analysis were further freeze-dried and milled in a planet-
ball mill.

TS contents were analyzed by means of dry combustion using an Eltra CS 2000 after combustion at 1250°C. The device was
previously calibrated with a certified coal standard and precision is better than ±0.02 %.

TRIS fractions were determined by a two-step sequential extraction of iron-monosulfides and pyrite (Fossing and Jørgensen,
1989). AVS was extracted by the reaction with 1 M HCl for 1 h under a continuous stream of di-nitrogen gas. The $H_2S$ released
was quantitatively precipitated as ZnS and then determined spectrophotometrically with a Specord 40 spectrophotometer
following the method of Cline (1969). The chromium-reducible fraction (CRS; essentially pyrite ($FeS_2$), was extracted with
hot acidic Cr(II)chloride solution. For $\delta^{34}S$ analysis in different TRIS fractions the ZnS was converted to $Ag_2S$ by addition of
0.1 M $AgNO_3$ solution with subsequent filtration, washing and drying of the $AgNO_3$ precipitate as described by (Böttcher and
Lepland, 2000). The amount and stable isotope composition of organically bond sulfur was measured from the washed and
dried solid residue after the Cr(II) extraction step via C-IRmMS following the approach of Passier (1999).

Reactive iron was extracted from freeze-dried sediments by the reaction with a 1 M HCl solution for 1 h (e.g., Canfield, 1989).
Iron was determined as $Fe^{2+}$ after reduction with hydroxylamine hydrochloride via spectrophotometry using ferrozine a



complexing agent following Stookey (1970). The extracted iron fraction consists of iron (III) oxyhydroxides and iron(II) monosulfides.

## 2.4 Microbial community analysis

Genomic DNA of 0.2-0.3 g sediment was extracted with the EurX Soil DNA Kit (Roboklon, Berlin, Germany) according to manufactory protocols. DNA concentrations were quantified with a Nanophotometer® P360 (Implen GmbH, München, DE) and Qubit® 2.0 Flurometer (Thermo Fisher Scientific, Darmstadt, Germany) according to the manufactory protocols.

The 16S rRNA gene for bacteria was amplified with the primer combination S-D-Bact-0341-b-S-17 and S-D-Bact-0785-a-A-21 (Herlemann et al., 2011). The 16S rRNA gene for archaea was amplified with the primer combination S-D-Arch-0349-a-S-17 and S-D-Arch-0786-a-A-20 (Takai and Horikoshi, 2000). The primers were labelled with unique combinations of barcodes. The PCR mix contained 1x PCR buffer (Tris•Cl, KCl, $(NH_4)_2SO_4$, 15 mM $MgCl_2$; pH 8.7) (QIAGEN, Hilden, Germany), 0.5 µM of each primer (Biomers, Ulm, Germany), 0.2 mM of each deoxynucleoside (Thermo Fisher Scientific, Darmstadt, Germany) and 0.025 U µl$^{-1}$ hot start polymerase (QIAGEN, Hilden, Germany). The thermocycler conditions were 95°C for 5 minutes (denaturation), followed by 40 cycles of 95°C for 1 minute (denaturation), 56°C for 45 seconds (annealing) and 72°C for 1 minute and 30 seconds (elongation), concluded with a final elongation step at 72°C for 10 minutes. PCR products were purified with a Hi Yield® Gel/PCR DNA fragment extraction kit (Süd-Laborbedarf, Gauting, Germany) according to the manufactory protocol. PCR products of three individual runs per sample were combined. PCR products of different samples were pooled in equimolar concentrations and compressed to a final volume 10 µl with a concentration of 200 ng µl$^{-1}$ in a vacuum centrifuge Concentrator Plus (Eppendorf, Hamburg, Germany). Individual samples were sequenced in duplicates.

The sequencing was performed on an Illumina MiSeq sequencer by the company GATC. The library was prepared with the MiSeq Reagent Kit V3 for 2x 300 bp paired-end reads according to the manufactory protocols. For better performance due to different sequencing length we used 15% PhiX control v3 library.

The quality of the sequences was checked using the fastqc tool (FastQC A Quality Control tool for High Throughput Sequence Data; http://www.bioinformatics.babraham.ac.uk/projects/fastqc/ by S. Andrews). Sequence raw reads were demultiplexed, and barcodes were removed with the CutAdapt tool (Martin, 2011). The subsequent steps included merging of reads using overlapping sequence regions (PEAR, Zhang et al. 2013), standardizing the nucleotide sequence orientation, and trimming and filtering of low quality sequences (Trimmomatic, Bolger et al. 2014). After quality filtering, chimera were removed by the ChimeraSlayer tool of the QIIME pipeline. Subsequently, sequences were clustered into operational taxonomic units (OTU) at a nucleotide cutoff level of 97% similarity and singeltons were automatically deleted. To reduce noise in the dataset sequences with relative abundances below 0.1% per sample were also removed. All archaeal libraries contained at least > 18.500 sequences, while bacterial libraries contained at least >12.500 sequences. OTUs were taxonomically assigned employing the GreenGenes database 13.05 (McDonald et al., 2012) using the QIIME pipeline (Caporaso et al., 2010).



Representative sequences of OTUs were checked for correct taxonomical classification by phylogenetic tree calculations in the ARB environment. Relative abundance of sequences related to known methanogens, anaerobic methanotrophs (ANME) and sulfate reducers were used to project microbial depth profiles. Sequences have been deposited at NCBI under the Bioproject PRJNA356778 with the sequence read archive accession numbers SRR5118134-SRR5118155 for bacterial and SRR5119428-SRR5119449 for archaeal sequences, respectively.

## 3. Results

### 3.1 Pore water geochemical patterns and pore water isotope composition

Substantial amounts of dissolved salts with EC maxima of up to 11.5 mS cm$^{-1}$ occurred at peat depths below 30 cmbsf. (cm below surface, Fig. 2a, Table A1) and corresponded with brackish pore water proportions of up to 60% (based on Baltic Sea salinity reported by (Feistel et al., 2010). Only at spot 1, with the greatest distance to the coastline, lower EC values (max. 3.4

mS cm$^{-1}$) indicated minor brackish pore water proportions (5-6%). At the other three spots, EC values were similar, i. e., exhibited no lateral salinity graduation along the remaining Baltic Sea-freshwater transect.

Vertical trends in pore water stable O isotope composition were similar for all spots and complementary to the salinity/EC patterns with an average upwards increase from 60 to 10 cmbsf. (Fig. 2b). The resulting salinity-$\delta^{18}$O relationship was negative (except for the low salinity gradient at freshwater spot 1) and thus inverse to the common salinity-$\delta^{18}$O trend characteristic for

Baltic coastal waters (Fig. 2c). This suggests that distribution patterns of salinity have formed independently from evaporation effects confined to the top pore water layers.

The pore water geochemistry in the peatland was increasingly diversified with depth: while the top 10 cmbsf. were comparatively homogenous across all spots, specific patterns evolving from diagenetic differences emerged primarily in deeper pore waters. Principal component analysis (Fig. 3) revealed the pore water geochemical composition below 10 cmbsf. to be

constrained by two major components that evolved in opposed lateral directions and, in concert, explained 90% of the variation in pore water composition: A distinct gradient associated with a depth increase of EC and the associated conservative ions (Cl$^{-}$, Na$^{+}$, Br$^{-}$) suggests a persistent brackish impact at spots 2, 3 and 4 (first principal component, explained 55% of the total variation). Only at spot 1, farthest away from the coastline, the EC increase with depth is minute. A second distinct lateral gradient is delineated by the concentrations of dissolved Fe, Mn, DIC, and Ca which occur in higher abundances at spot 1 and

2 closest to the upstream terrestrial catchment boundary (second principal component, explained 35% of the total variation). Such a lateral shift in pore water geochemistry is probably related to the supply of mineral solutes from terrestrial inflow. In this regard, the pore water composition of spot 2 unites the elevated supply in mineral compounds from terrestrial inflow with persisting remnants of former brackish impact.

### 3.2 Sulfur speciation, S isotope patterns and sulfate reducing communities

We found distinct differences in the S biogeochemical patterns across spots indicating different sulfate supply and transformation processes along the terrestrial-brackish continuum. In the following, we structured the results spot-wise





according to the specific S regime and discuss first spot 1 (low solid sulfur and low sulfate), then spots 3 and 4 (high solid sulfur and low sulfate) and finally spot 2 (high solid sulfur and partially high sulfate concentrations).

### 3.2.1 Spot 1

Spot 1 characterized by low salinities and implied mineral inflow from the near freshwater catchment, exhibited the lowest sulfate concentrations of $\leq 0.3$ mM and $H_2S$ concentrations were below detection limit (~1 µM, Fig. 4). Sulfate made up only a small proportion of the total dissolved S pool, thereby indicating a higher abundance of a non-specified dissolved S fraction, probably composed of dissolved organic S, polysulfides, and S intermediates.

Also, the abundance of solid S was lowest at spot 1 ($\leq 0.7$ %dwt total S). Among solid S compounds, organic-bond S constituted

the dominant solid S fraction (0.2 to 1.6 %dwt) with relatively stable $\delta^{34}S$ ratios (+8.1 and +9.8‰). Pyrite contents (measured as CRS) were low despite of abundant pore water Fe and available solid iron (Fig. 5). Only at spot 1, we found a low though consistent abundance of iron mono-sulfides (0.05-0.08 %dwt, measured as AVS). Biogeochemical turnover processes here might operate under sulfate-limited conditions resulting in lower sedimentary S contents and accumulation of iron monosulfides.

In correspondence with the low sulfate contents, no sulfate reducing bacteria were found at spot 1.

### 3.2.2 Spots 3 and 4

Despite the persisting brackish impact found in the deeper pore waters of spots 3 and 4 closest to the Baltic Sea, we found only low abundances of pore water sulfate in the top 5 cmbsf. (0.3-0.5 mM), concentrations below detection limit (~0.001 mM) in a depth range of 10 to about 30 cmbsf., and moderate $SO_4^{2-}$ levels below 30 cmbsf. (0.1-1 mM). $H_2S$ abundance was essentially

restricted to depth at spot 3 (up to 0.4 mM).

Low porewater sulfate concentrations prevented $\delta^{34}S$ measurements at the majority of the data points. However, the single $\delta^{34}S$ value of +86.4‰ measured at 60 cmbsf. of spot 3 (Fig. 6a) indicates a remarkable $^{34}S$ enrichment in relation to Baltic Sea water $SO_4^{2-}$ (+21‰; Böttcher et al., 2007). Sulfur isotope fractionation to this extent is likely to result from a superposition of enzymatic kinetic fractionation associated with a reservoir effect and constitutes striking isotopic evidence for the exhaustion

of the brackish sulfate pool by intense DSR (Hartmann and Nielsen, 2012). Despite the missing isotopic evidence, it is likely, that the low sulfate concentrations at the remaining depth sections of spot 3 and along the depth profile of spot 4 result from the same intense sulfate reduction processes.

At the depths of potentially high DSR, we measured high amounts of total solid S (TS up to 4% dwt). Organic-bond S constituted the dominant solid S fraction (0.2 to 1.6 %dwt) in these spots. Pyrite was less abundant (max 0.3%dwt) and

exhibited a wide range of $^{34}S$ ratios ranging from -15 to +11‰. As pyrite $\delta^{34}S$ ratios essentially reflect the isotopic signature of the microbially derived sulfide pool (Butler et al., 2004; Price and Shieh, 1979), the found variation in pyrite $\delta^{34}S$ ratios reflected a differentiated reservoir effect that amplifies the enzymatic kinetic fractionation of DSR and varies in response to the openness of the system (i. e. connectivity to the sea).





In correspondence with the exhaustion of the brackish sulfate pool, the relative abundance of SRB was generally small (<5%)

and most likely substrate-limited. SRB were from the *Deltaproteobacteria* class and the *Thermodesulfovibrionaceae* genus of the *Nitrospirae* phylum. With 40% relative abundance, *Chloroflexi* of the class *Dehalococcoidetes* represented the dominating bacterial group at the 1 mM $SO_4^{2-}$ concentration depth of spot 3.

### 3.2.3    Spot 2

At spot 2 - the interface between brackish impact and mineral inflow from the freshwater catchment - we found a sharp rise in

$SO_4^{2-}$ concentration from <0.3 mM at the top 20 cm up to 33 mM at 60 cmbsf. The latter exceeded the quantities expected from marine supply (Feistel et al., 2010; Kwiecinski, 1965) by a factor of 8. The pronounced concentration gradient at spot 2 was associated with a remarkable variation in the stable isotope composition showing a downcore decrease in $\delta^{34}S$-$SO_4^{2-}$ from +82.9 to +22.7‰ and a decrease in $\delta^{18}O$-$SO_4^{2-}$ from +30 to +11‰ (Fig. 6a). $\delta^{34}S$ values >+80‰ at 30 cmbsf. of spot 2 suggest the brackish sulfate pool in the top pore waters to be microbially exhausted under the same reservoir effect as in spots 3 and

4. The $\delta^{18}O$ and $\delta^{34}S$ ratios of excess $SO_4^{2-}$ in 60 cmbsf. ($\delta^{34}S$: +22.7‰, $\delta^{18}O$: +11.4‰) corresponded well with modern day seawater $SO_4^{2-}$ ($\delta^{34}S$: +21‰, $\delta^{18}O$: +9‰, Böttcher et al., 2007). Altogether, the sharp sulfate concentration and isotope gradients at spot 2 could demonstrate the entire spectrum of sulfate speciation from the persistence of a marine sulfate reservoir at 60 cmbsf. towards progressing sulfate depletion in the upper peat layers.

To test this hypothesis, we applied a closed-system (Rayleigh-type) model (Eq. (1), Mariotti et al., 1981) to the data from spot

2 and gained an estimate for the $\delta^{34}S$ ratios of the initial $SO_4^{2-}$ reservoir ($\delta^{34}S_{SO_4^{2-}initial}$) and the kinetic isotope enrichment factor ε:

$$\delta^{34}S_{SO_{4,depth}^{2-}} - \delta^{34}S_{SO_{4,initial}^{2-}} = \varepsilon\ln(fSO_{4,depth}^{2-}) \tag{1}$$

Here $\delta^{34}S_{SO_4^{2-}depth}$ represents the S isotope values measured in specific depths of spot 2, and $fSO_4^{2-}depth$ constitutes the fraction of remaining pore water $SO_4^{2-}$ in relation to the initial sulfate reservoir (33 mM $SO_4^{2-}$, measured in 60 cmbsf at spot 2). The fit

through four data points ($R^2$: 0.99; p>0.05) revealed the $\delta^{34}S$ ratios of the initial $SO_4^{2-}$ reservoir (+24‰) to be close to the $^{34}S$ signature of the Baltic Sea (Fig. 6b). The isotopic offset is within the uncertainty of the estimate. The isotope enrichment factor ε was estimated to be -27‰ which is within the range reported for DSR in laboratory studies with pure cultures (Canfield, 2001; Kaplan and Rittenberg, 1964; Sim et al., 2011) and in the field (Böttcher et al., 1998; Habicht and Canfield, 1997).

The pronounced sulfate distribution patterns at spot 2 went along with the highest amounts of pyrite (0.48-1.36 %dwt.) that

exceeded the amounts of organic-bond S (0.38-1.26 %dwt.). The patterns in pyrite $\delta^{34}S$ ratios did not correspond with the vertical trend in sulfate availability. Instead $\delta^{34}S$ values were lowest in 20 cmbsf. (-15‰) and stabilized around +2‰ below. Interestingly, at peak sulfate supply of spot 2, the relative abundance of *Deltaproteobacteria* did not exceed 5%. Instead, the SRB community at depth was dominated by the *Thermodesulfovibrionaceae* genus which contributed up to 21% of all bacterial 16S rRNA sequences. Likewise with spot 3, *Chloroflexi* of the class *Dehalococcoidetes* represented also the dominating

bacterial group at depth of spot 2.





### 3.3 Dissolved methane concentrations, isotopic signature and methanogenic communities

Measured pore water $CH_4$ concentrations were up to 0.7 mM with equivocal vertical patterns across spots (Fig. 7a), reflecting the methane-specific spatial variability that evolves from small-scale heterogeneity in production and consumption processes and from ebullitive release events (Chanton et al., 1989; Whalen, 2005). The isotope composition of $CH_4$ (Fig. 7b) and DIC

(Fig. 7c) provided a clearer (and probably more robust) indication for patterns of methanogenesis and methanotrophy: High $\delta^{13}$C-DIC ratios up to +4.2‰ suggest intense methanogenic (i. e. $^{13}$C-DIC fractionating) processes in 20-40 cmbsf, whereas DIC on top was comparatively depleted in $^{13}$C as characteristic for methane oxidation in the aerated surface layers. $\delta^{13}$C-DIC ratios below 40 cmbsf. converged towards the isotopic signature of bulk organic C (-26‰).

At spot 2, we found the most pronounced downward drop in $\delta^{13}$C-DIC ratios with a minimum of -23.9‰ in 60 cmbsf. This

pattern coincided with a consistent downward decrease in $\delta^{13}$C-$CH_4$ ratios from -57 to -68‰ and suggests that methanogenesis operates under higher $^{13}$C fractionation associated with thermodynamically less favorable conditions at the bottom of spot 2. $\delta$D ratios of methane did not exhibit a concurrent increase but varied unrelated to $\delta^{13}$C-$CH_4$ ratios in a range between -333 and -275‰. Based on the C and D isotopic ratio threshold raised by Whiticar (1986), acetate fermentation revealed to be the dominant methane production pathway in our study site (Fig. 8). A concurrent rise in both $\delta$D- and $\delta^{13}$C-$CH_4$ ratios at depth

of spot 1 suggests a shift towards dominating $CO_2$ reduction and/or an increase in methanotrophy.

Together with high $\delta^{13}$C-DIC ratios in the upper parts of the peat, 16S rRNA sequences related to methanogens (Fig. 7d) provided further evidence for intensive methane production. At spot 2, we found the largest divergence with 90% methanogen-related sequences at the surface while in deeper regions (10-50 cmbsf.) less than 7% of the archaeal domain could be attributed to methanogens. Surprisingly, at 60 cmbsf. of spot 2, methanogen percentages increased abruptly up to 41% despite of high

relative abundances of SRB. Spot 1 exhibited the lowest methanogen proportions which decreased from 21% at the top down to 1% in 50 cmbsf.

The methanogen community was mostly dominated by *Methanoseata*, an obligate acetotrophic archaea genus that thrives in terrestrial organic-rich environments. *Methanosaeta* proportion usually scaled with the methanogen percentage, and contributed 70-100% to the methanogenic community.

The elucidation of in situ methanogenic pathways from the isotopic composition of $CH_4$ can be blurred by overlapping fractionation factors or the isotope effect of methanotrophy. However, the phylogenetic structure of the methanogenic community provided unequivocal evidence for acetate fermentation as the prevailing methanogenic pathway in most of the peatland.

Sequences related to aerobic methanotrophs of the genus *Methylosinus* were only found at 30 cmbsf. in spot 4 representing

approximately 1.5% of all bacterial sequences (data not shown). Aerobic methanotrophs were underrepresented in our dataset. Consistent with the concurrent depth increase in $\delta^{13}$C-$CH_4$ and $\delta$D-$CH_4$, spot 1 (Fig. 8), situated at the fringe of the freshwater catchment, exhibited high abundances of anaerobic methanotrophs of the ANME-2d clade, that are so far implicated to use $NO_3^-$ (Raghoebarsing et al., 2006) and/or Fe(III) (Ettwig et al., 2016) as electron acceptor.



## 4. Discussion

### 4.1 Pore water biogeochemical patterns

Overall, the pore water geochemistry of the Hütelmoor is characterized by lateral legacy effects preserved in 20 to 30 cm depth that reflect the brackish/terrestrial continuum and an overlying recent layer representing the prevalent freshwater regime induced by rewetting.

Despite a continuous ground water inflow from the forested catchment (Miegel et al., 2016), relics of former brackish and
mineral terrestrial inflow are preserved in the deeper layers of the relatively shallow peat body. This is exemplified by high pore water EC values that exceeded those reported directly after the last brackish water intrusion event in 1995 (Bohne and Bohne, 2008). In fact, discharge within the peatland is channeled through rapid flow in the drainage ditches while water movement within the interstitial peat body seems to be mostly restricted to vertical exchange processes (evaporation, precipitation) with minor lateral flow (Selle et al., 2016). Therefore, we assume that drainage-induced hydrological alterations
reinforce the segregation of the peat pore matrix from subsurface lateral exchange. This allows for the preservation of residual signals in deeper pore waters and confines contemporary biogeochemical transformation processes to the recycling of autochthonous matter. The newly established freshwater layer on top overprints lateral differences along the brackish/fresh continuum and determines the upper pore water geochemistry in the entire peatland.

### 4.2 Sulfur transformation

Along the entire brackish/terrestrial transect, virtually no sulfate was abundant in the newly developed fresh pore water layer at the top 20 cm. However, distinct differences in sulfur speciation across spots were preserved below 20 cmbsf. and seemed to reflect the gradual exposure to former brackish intrusion and terrestrial inflow.

Spot 1 appeared to be virtually un-affected by any brackish impact with biogeochemical turnover processes operating under sulfate-limited conditions. Low sedimentary S contents and the accumulation of iron monosulfides as representative for
freshwater environments are strong points for this conclusion.

Also at spots 3 and 4, contemporary biogeochemical processes essentially operated under sulfate-limited conditions although these areas had been exposed to flooding from the nearby Baltic Sea. High sedimentary S concentrations in conjunction with the $^{34}$S composition of the remaining sulfate suggest that the brackish sulfate reservoir has been essentially exhausted through DSR with the produced sulfide being either incorporated as diagenetically derived S in organic compounds or precipitated as
$^{34}$S-enriched pyrite minerals (Brown and MacQueen, 1985; Hartmann and Nielsen, 2012). Hence, if dyking of coastal wetlands prevents the replenishment of the brackish sulfate reservoir, the latter can be almost completely consumed through DSR as has been demonstrated by the Rayleigh distillation model. The rapid exhaustion of the brackish sulfate reservoir is likely to be reinforced in coastal peatlands where vast amounts of C compounds constitute an extensive electron donor supply for DSR.

Prevalent sulfate-limitation at spots 1, 3 and 4 was reflected by the virtual absence of the sulfate reducing microbial community.
Interestingly, minor remnants of the brackish sulfate pool (1 mM $SO_4^{2-}$) at depth of spot 3 were associated with 40% relative abundance of *Chloroflexi* of the class *Dehalococcoidetes*. Genomes of this group in marine sediments have been shown to





code for dsrAB genes (Wasmund et al., 2016). Through their ability to reduce sulfite they may be involved in S redox cycling. Indeed, further research is required to better establish their function in the S cycle.

S geochemistry at spot 2, which unites the effects of brackish water intrusion with mineral inflow of terrestrial origin, differed
substantially from the other spots with remarkably high sulfate concentrations (33mM) at depth. The mineral impact from terrestrial inflow was not only reflected by high concentrations of dissolved constituents (Fe, DIC, Mg, Ca, Mn) but also by high contents of labile iron minerals and dissolved ferrous iron. Interactions with poorly-ordered ferric hydroxides can supply Fe(III) as competitive electron acceptor next to sulfate (Postma and Jakobsen, 1996) and may, therefore, inhibit the efficient microbial reduction of the brackish sulfate reservoir. Amorphous ferric hydroxides effectively suppressed DSR in a recently
rewetted Baltic coastal wetland (Virtanen et al., 2014). In our study, high contents of labile iron minerals and dissolved ferrous iron at depth of spot 2, coincided with a high abundance of *Thermodesulfovibrionaceae* at concurrently minor occurrence of *Deltaproteobacteria.* Recent in vitro experiments suggest *Thermodesulfovibrionaceae* can utilize ferric iron as electron acceptor next to sulfate (Fortney et al., 2016). Indeed, the demonstration of Fe(III) reduction by *Thermodesulfovibrionaceae* under in situ conditions is currently still pending. But high contents of labile iron minerals, the remarkable accumulation of
pore water iron, and the absence of typical iron reducers (*Geobacteraceae, Peptococcaceae, Shewanellaceae, Desulfovibrionaceae, Pelobacteraceae*) could suggest *Thermodesulfovibrionaceae* to prefer Fe(III) as electron acceptor over sulfate. The unique $SO_4^{2-}$ concentration patterns at spot 2 may, thus, be attributed to the inhibited microbial consumption of the brackish sulfate reservoir caused by the delivery of alternative electron acceptors from the nearby freshwater catchment.

Altogether, our results demonstrate the fate of the brackish sulfate reservoir in coastal wetlands under closed system conditions
caused by dyking. Microbial transformation processes have decoupled the sulfate distribution patterns from the relic brackish impact delineated through conservative tracer ions and have caused marked differences in contemporary sulfate biogeochemistry: On one hand, the brackish sulfate reservoir was efficiently exhausted by DSR, whereas on the other hand, the preferential consumption of competitive electron acceptors from terrestrial origin allowed for the local accumulation of large sulfate concentrations. Indeed, these relic signals of brackish-terrestrial intermixing are constrained to the deeper pore
water regions below 30 cmbsf. as recent rewetting measures established a homogeneous freshwater regime in the top layers of the entire peatland.

### 4.3 Methane production and consumption

$\delta^{13}$C-DIC ratios and a thriving methanogenic community indicate the establishment of distinct methane production zones in the recently formed freshwater layer across the entire peatland. Interestingly, the methanogen community was dominated by
*Methanoseata*, an obligate acetotrophic genus of archaea typical of terrestrial organic-rich environments, indicating the prevalent freshwater characteristics of the newly formed pore water layer. Indeed, thermodynamically favorable methanogenic conditions were confined to the top layers since isotopic evidence and archaeal distribution patterns indicate a downward shift towards non-fractionating metabolic processes (Barker, 1936; Lapham et al., 1999) at the bottom. This vertical transition was



most pronounced at spot 2, probably indicating a potential suppression of methanogenesis by high concentrations of sulfate

and labile ferric iron compounds at depth.

Surprisingly, we observed mutual coexistence of SRB (22% of all bacterial sequences) and methanogens (>40% of all archaeal sequences) at high $SO_4^{2-}$-concentrations (32.8 mM) in 60 cmbsf. at spot 2. Simultaneous methanogenesis and DSR have been reported under the abundance of methanol, trimethylamine or methionine as methanogenic precursors (Oremland and Polcin, 1982). However, the concurrent high abundance of *Methanosaeta* (30%) at depth of spot 2 suggests competitive consumption

of acetate by both SRB and methanogens. Although Liebner et al. (2015) emphasized the relevance of community structure with regards to prevailing methanogenic pathways, total abundance data could potentially yield more insights to this issue.

Sequences related to aerobic methanotrophs of the genus *Methylosinus* were only found at 30 cmbsf. in spot 4 representing approximately 1.5% of all bacterial sequences (data not shown). The phenomenon of a lagged re/establishment of methanotrophs in comparison to methanogens after rewetting in this particular peatland is addressed in another publication

(Wen et al., in review, 2018).

Despite the overlap of modeled methane production and sulfate reduction zones, we couldn't find evidence for the syntrophic consortium of anaerobic methanotrophs (ANME) and sulfate reducers that is commonly associated with AOM-SR in marine environments (Boetius et al., 2000). However, we cannot exclude that AOM-SR is driven by archaea that are so far not known for this function. One potential candidate phylum is the *Bathyarchaota* that have been shown to encode an untypical version

of the functional gene for methane production and consumption (methyl co-enzyme M reductase subunit A, *mcrA*) (Evans et al., 2015). These archaea dominated spot 2 with 48-97% relative sequence abundance of the archaeal community between 10 and 60 cm (data not shown). While we cannot supply microbial evidence for AOM-SR, high abundances of anaerobic methanotrophs of the ANME-2d clade at spot 1 suggest anaerobic methane oxidation coupled to electron acceptors of terrestrial origin. Methanotrophs of the ANME-2d clade are so far known to utilize $NO_3^-$ (Raghoebarsing et al., 2006) and ferric iron

(Ettwig et al., 2016) as electron acceptors, both of which were abundant at the respective spot. This observation is further supported by the trend in $\delta^{13}C$-$CH_4$ and $\delta D$-$CH_4$ which potentially indicates a downward increase in methanotrophy at spot 1.

Our results demonstrate how rewetting of a coastal peatland established a distinct freshwater regime in the upper pore water layers, which, in conjunction with prevalent anaerobic conditions and a vast stock of labile C compounds, offers favorable conditions for intense methane production and explains the high methane emissions reported in (Hahn et al., 2015) and

Koebsch et al. (2015). As intense methane production was confined to the upper pore water layers in the entire peatland, it did not interfere with high sulfate concentrations locally preserved as legacy of former brackish impact in the bottom. Instead, isotopic and microbial evidence suggested mineral compounds of terrestrial origin to constitute an electron acceptor for anaerobic methane oxidation, which is an often neglected - though important process in freshwater environments (Segarra et al., 2015). Our results indicate that this process occurs also in disturbed coastal peatlands. Indeed, the quantitative effects of

anaerobic methane consumption on methane emissions in coastal and/or rewetted peatlands need to be addressed in future studies.





## 5.  Conclusions

In this study, we investigated the biogeochemical and hydrological mechanisms that turn disturbed and remediated coastal peatlands into strong methane sources. Our study highlights how human intervention overrides the sulfate-related methane
emission suppression processes that constitute a natural greenhouse gas mitigation mechanism in coastal environments. Hence, the climate effect of disturbed and remediated coastal wetlands cannot simply be derived by analogy with their natural counterparts. Instead, human alterations form new transient systems where relic brackish signals intermingle with recent freshwater impacts. The evolving biogeochemical patterns overprint naturally established gradients formed, for instance by the distance to the coastline. In particular, the decoupling of sulfate abundance from salinity is of high practical relevance for
greenhouse gas inventories that establish methane emission factors based on the empirical relation to salinity as easily accessible proxy for sulfate concentrations.

Coastal environments are subject to particular pressure by high population density while at the same time their potential as coastal buffer zones is moving more and more into the focus of policy makers and land managers. From a greenhouse gas perspective, the exposure of embanked wetlands to natural coastal dynamics would literally open the floodgates for a
replenishment of the marine sulfate pool and constitute an efficient measure to reduce methane emissions. However, in practice, this option has to be weighed against concurrent land use aspects.

## 6.  Data availability

Geochemical data are represented within this manuscript in the appendix (Table A1). Sequences have been deposited at NCBI under the Bioproject PRJNA356778 with the sequence read archive accession numbers SRR5118134-SRR5118155 for
bacterial and SRR5119428-SRR5119449 for archaeal sequences, respectively.





## 7.  Appendices

**Table A1 Molar concentrations and dry weight ratios of dissolved and solid species**

| Station | Depth cm | Sal ppt | EC mS cm⁻¹ | Cl mM | Br µM | Na mM | TS mM | SO₄²⁻ mM | H₂S µM | TS %dwt | CRS %dwt | AVS %dwt | orgS %dwt | CH₄ µM | DIC mM |
|---|---|---|---|---|---|---|---|---|---|---|---|---|---|---|---|
| | 0 | 0.7 | 1.8 | 11.5 | 19.9 | 9.6 | 0 | 0 | 1 | 0.3 | 0.1 | 0.1 | 0.2 | 144 | 5.4 |
| | 5 | 0.7 | 1.8 | 12.6 | 19.9 | 10.7 | 0 | 0 | 0 | 0.3 | 0.1 | 0.1 | 0.2 | 312 | 6.2 |
| | 10 | 1.0 | 2.4 | 14.6 | 19.1 | 10.7 | 0 | 0 | 3 | 0.3 | 0.1 | 0.1 | 0.3 | 234 | 7.5 |
| 1 | 20 | 1.4 | 2.9 | 11.0 | 25.6 | 10.5 | 0 | 0 | 1 | 0.3 | 0.1 | 0.1 | 0.5 | 109 | 21.7 |
| | 30 | 1.6 | 3.4 | 12.5 | 31.9 | 14.1 | 0 | 0 | 1 | 0.3 | 0.1 | 0.0 | 0.5 | 143 | 25.3 |
| | 40 | 1.7 | 3.4 | 11.4 | 31.3 | 13.7 | 0 | 0 | 2 | 0.5 | 0.1 | 0.1 | 0.4 | 178 | 26.7 |
| | 50 | 1.5 | 3.2 | 12.0 | 38.1 | 13.5 | 0 | 0 | 0 | 0.7 | 0.1 | 0.1 | 0.3 | 101 | 21.8 |
| | 0 | 1.4 | 3.0 | 19.3 | 37.0 | 18.2 | 0 | 0 | 0 | 1.3 | 0.5 | 0.1 | 0.5 | 462 | 8.9 |
| | 5 | 1.2 | 2.6 | 23.3 | 39.0 | 17.8 | 0 | 0 | 1 | 1.8 | 0.5 | 0.1 | 0.8 | 344 | 8.4 |
| | 10 | 3.0 | 5.7 | 37.9 | 46.5 | 32.6 | 1 | 0 | 6 | 2.3 | 0.5 | 0.0 | 1.0 | 56 | 17.3 |
| 2 | 20 | 4.0 | 7.3 | 48.3 | 82.1 | 41.4 | 1 | 0 | 7 | 2.3 | 0.7 | 0.0 | 1.3 | 82 | 20.8 |
| | 30 | 5.4 | 9.7 | 63.7 | 99.8 | 56.5 | 4 | 4 | 5 | 3.4 | 0.8 | 0.0 | 1.2 | 643 | 28.8 |
| | 40 | 5.4 | 9.7 | 64.9 | 125.3 | 64.3 | 19 | 17 | 34 | 1.7 | 1.0 | 0.0 | 1.1 | 197 | 15.5 |
| | 50 | 5.5 | 9.9 | 67.8 | 129.5 | 61.7 | 18 | 19 | 61 | 4.0 | 1.2 | 0.0 | 0.9 | 128 | 17.1 |
| | 60 | 6.5 | 11.5 | 75.5 | 85.8 | 63.9 | 33 | 33 | 274 | 0.5 | 1.4 | 0.0 | 0.4 | 139 | 12.8 |
| | 0 | 1.4 | 2.9 | 22.2 | 151.6 | 19.6 | 0 | 0 | 0 | 0.9 | 0.2 | 0.0 | 0.3 | 231 | 4.4 |
| | 5 | 1.4 | 3.0 | 22.4 | 49.8 | 20.9 | 0 | 0 | 1 | 1.1 | 0.2 | 0.0 | 0.3 | 193 | 4.9 |
| | 10 | 1.9 | 3.8 | 28.6 | 50.9 | 28.1 | 0 | 0 | 21 | 1.3 | 0.2 | 0.0 | 0.3 | 486 | 6.1 |
| 3 | 20 | 3.7 | 6.8 | 54.5 | 64.9 | 48.3 | 1 | 0 | 53 | 1.2 | 0.2 | 0.0 | 0.2 | 420 | 5.7 |
| | 30 | 4.7 | 8.6 | 69.4 | 122.9 | 58.7 | 1 | 0 | 38 | 1.6 | 0.2 | 0.0 | 1.0 | 81 | 4.1 |
| | 40 | 5.4 | 9.6 | 87.2 | 156.3 | 55.7 | 0 | 0 | 25 | 2.4 | 0.2 | 0.0 | 1.6 | 122 | 4.1 |
| | 50 | 5.7 | 10.2 | 92.8 | 168.5 | 77.0 | 1 | 0 | 187 | 2.9 | 0.2 | 0.0 | 1.0 | 13 | 3.6 |
| | 60 | 5.2 | 9.4 | 77.6 | 181.6 | 70.9 | 2 | 1 | 347 | 3.5 | 0.2 | 0.0 | 1.0 | 89 | 6.3 |
| | 0 | 1.4 | 2.9 | 20.5 | 159.4 | 19.2 | 0 | 0 | 1 | 1.3 | 0.3 | 0.0 | 0.6 | 254 | 4.2 |
| | 5 | 1.2 | 2.7 | 22.6 | 49.4 | 19.8 | 0 | 0 | 0 | 1.0 | 0.2 | 0.0 | 0.6 | 127 | 4.0 |
| | 10 | 2.7 | 5.2 | 37.7 | 48.4 | 33.1 | 1 | 0 | 7 | 0.7 | 0.2 | 0.0 | 0.6 | 48 | 8.6 |
| 4 | 20 | 3.2 | 6.1 | 52.3 | 84.9 | 44.3 | 1 | 0 | 5 | 0.8 | 0.2 | 0.0 | 0.4 | 49 | 6.6 |
| | 30 | 4.5 | 8.1 | 69.4 | 99.3 | 55.2 | 1 | 1 | 2 | 1.5 | 0.2 | 0.0 | 0.3 | 292 | 11.6 |
| | 40 | 4.5 | 8.2 | 73.5 | 126.1 | 50.4 | 0 | 0 | 33 | 0.2 | 0.2 | 0.0 | 0.4 | 430 | 11.3 |



## 8. Author contributions

FK and MB have formulated the research question and planned the study design. FK acquired funding. FK, GJ, MK, MW and SK collected the samples. MB, SL, AS, MG, TS and SK provided resources and lab instrumentation for sample analysis. FK, AS, IS, MK, GJ, SK and JW conducted the geochemical analyses. MW, SL and VU conducted the microbial sequencing analysis. BL validated the results. FK visualized the data and prepared the original draft with contributions from all coauthors.

## 9. Competing interests

The authors declare that they have no conflict of interest

## 10. Acknowledgements

This work was supported by the DFG Research Training Group *BALTIC TRANSCOAST* (grant DFG GRK 2000) and the Helmholtz *Terrestrial Environmental Observatories* (*TERENO*) Network. This is *BALTIC TRANSCOAST* publication number GRK2000/00X. FK was supported by the Helmholtz Association of German Research Centers through the Helmholtz Postdoc

Programme (grant PD-129) and the Helmholtz Climate Initiative *REKLIM* (Regional Climate Change). TS and SL are each supported by a Helmholtz Young Investigators Group (grant VH-NG-821 and VH-NG-919). Biogeochemical and stable isotope work was further supported by the Leibniz Institute for Baltic Sea Research (IOW). We wish to express our gratitude to L. Kretzschmann, A. Saborowski, and S. Strunk for their commitment to field work under tough conditions. B. Juhls and S. Strunk have helped with map creation. The study would not have been possible without the laboratory and bioinformatics

support by A. Gottsche, A. Saborowski, L. Kretzschmann, A. Köhler, B. Plessen, V. Winde, U. Günther, F. Horn, X.Wen, and H. Baschek.

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





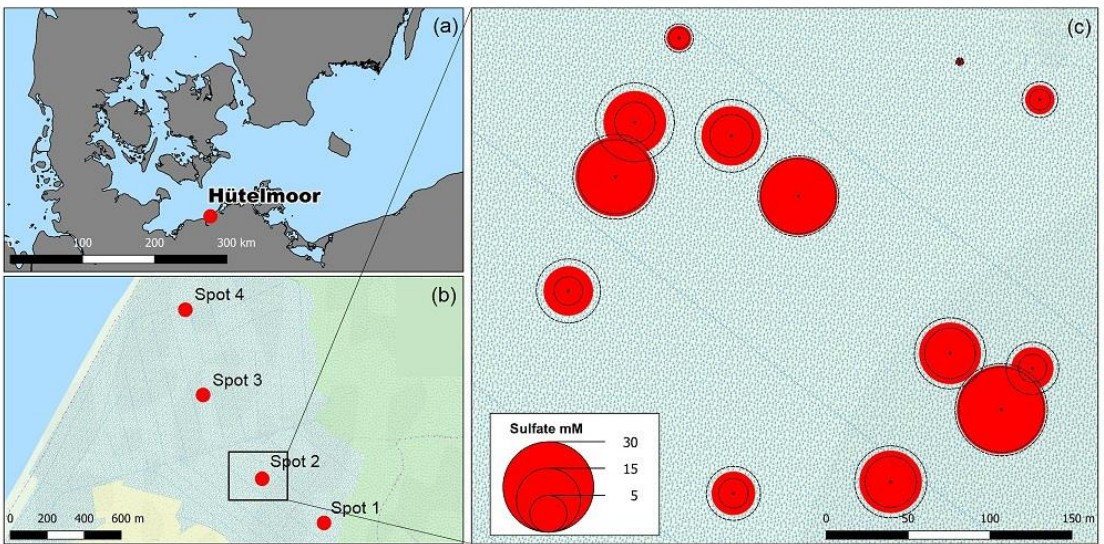

**Figure 1: (a) The study site Hütelmoor is located directly at the south-western Baltic coast at an altitude between -0.2 and +0.2 m above sea level. In its pristine state, the site was exposed to episodic brackish water intrusion by storm surges. (b) Profiles of sediments and pore waters were taken along a transect with 300-1,500 m distance to the coast line. Deviations of the transect from the straight normal to the Baltic coast line arose due to the restricted accessibility of the site. (c) A former study located close to spot 2 in the center of the current sampling transect revealed high pore water sulfate concentrations in 30-60 cm below surface with annual means up to 24±3 mM (red circles indicate annual means while dashed circle lines represent the standard deviation over the year). Map data copyrighted OpenStreetMap contributors and available from http://www.openstreetmap.org.**





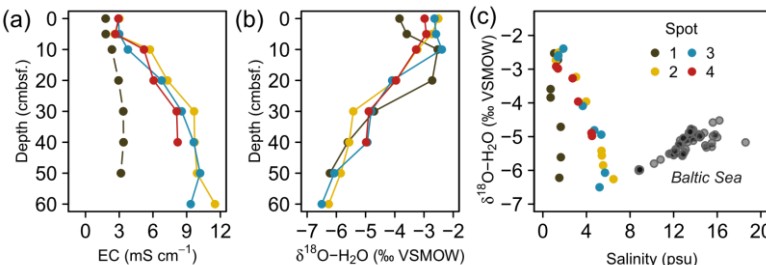

**Figure 2: Depth distributions (a and b) and scatter plots (c) of electrical conductivity (EC) respectively. salinity and pore water O isotope composition. Filled dots in Fig. 2c represent a common positive δ¹⁸O-H₂O vs. salinity relationship derived from a sampling campaign of Baltic Sea surface water (unpublished).**






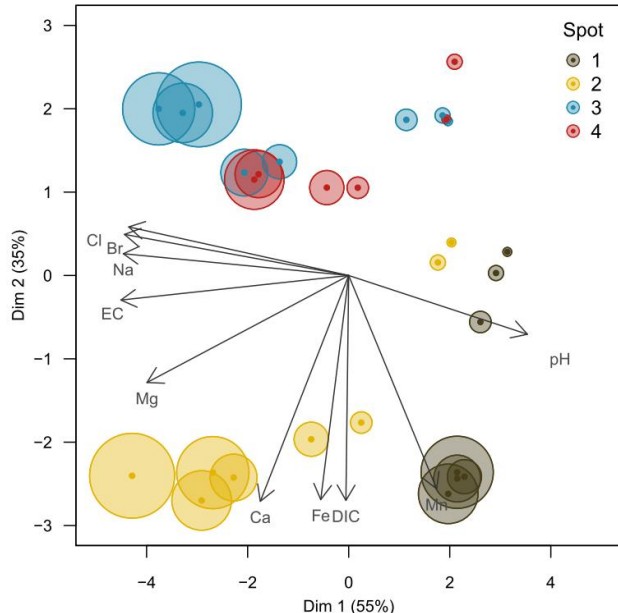

**Figure 3: Principal component biplot of pore water geochemical patterns within the peatland. Different colors indicate different sampling locations within the brackish-freshwater continuum with spot 1 closest to the freshwater catchment and spot 4 closest to**
**the Baltic Sea. The size of the data points scales with sampling depth (smallest points indicate surface patterns, largest points indicate pore water composition in 60 cm depth.**





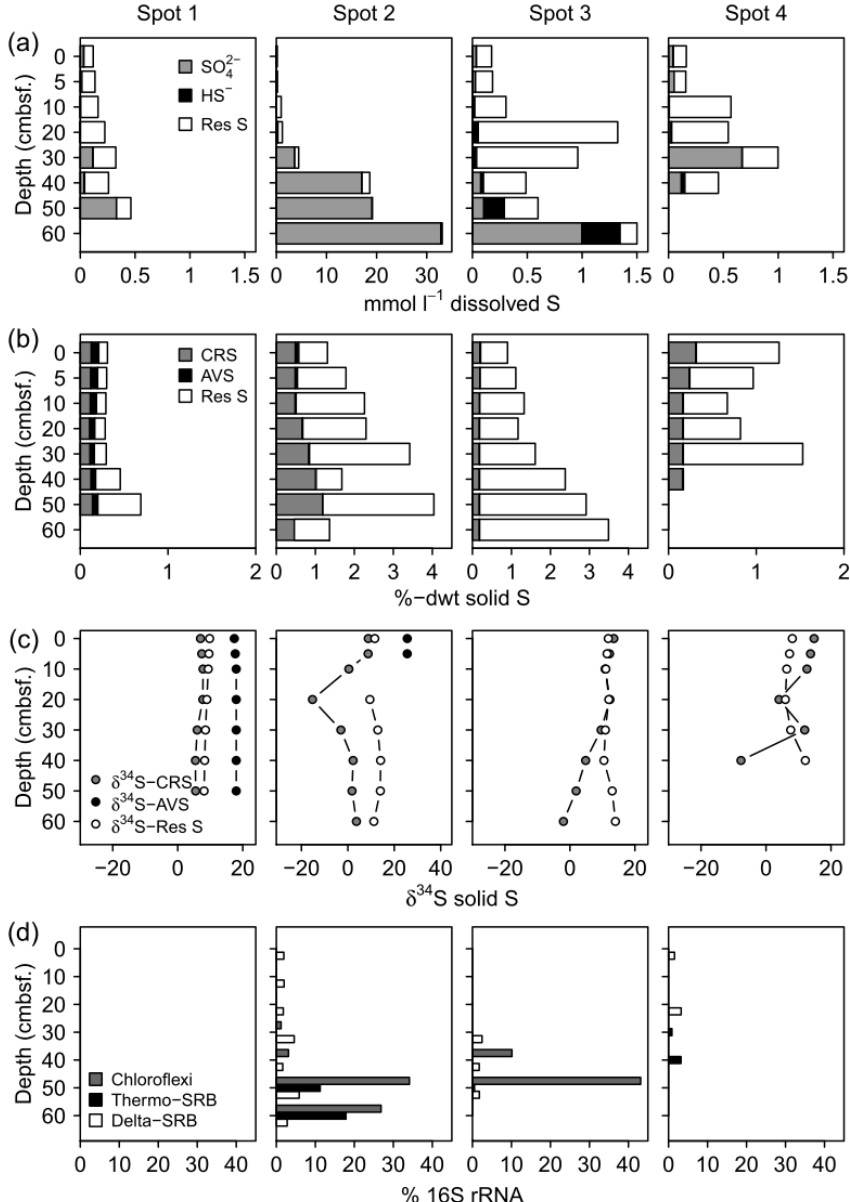

**Figure 4: Speciation of dissolved (a) and solid (b) S compounds, S isotope composition of solid S compounds (c), and average relative abundances of sulfate reducing bacteria (SRB, d). Sufficient $SO_4^{2-}$ for $\delta^{34}S$ and $\delta^{18}O$ ratio analysis was only available at the bottom of spot 2 and 3 and are displayed in Fig. 6a. Residuals for dissolved and solid S refer to non-specified S fractions evolving from the difference between a total S quantity and specified S compounds. The dissolved residual S fraction is most likely composed of dissolved organic S, polysulfides, and S intermediates. The solid residual S fraction is suggested to present primarily organic-bond S. Specified solid S fractions include iron mono-sulfide operationally defined as acid volatile sulfur (AVS) and pyrite extracted as chromium-reducible sulfur (CRS). $\delta^{34}S$ at AVS could only be measured at spot 1 and the top of spot 2. SRB were extracted from two replicates of 16S rRNA bacterial community sequencing and are assigned to the Deltaproteobacteria (Delta-SRB) and the Nitrospirae phylum (Genus Thermodesulfovibrionaceae – Thermo-SRB). Chloroflexi Dehalococcoides (Chloroflexi) have not been assigned to SRB in the classical sense, however, they could be potentially involved in S metabolism (Wasmund et al., 2016). Note different x axis scales.**



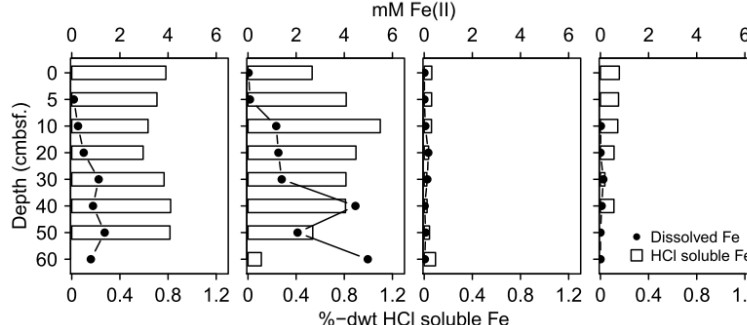


**Figure 5: Mobile Fe species. Available solid iron was extracted as HCl soluble iron from the sediment matrix and is composed of iron mono-sulfide and non-sulfidized ferric Fe.**




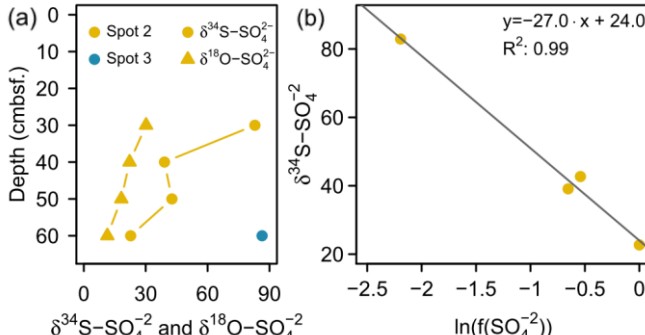

**Figure 6: (a) S and O isotope composition of sulfate. Sufficient $SO_4^{2-}$ for $\delta^{34}S$ and $\delta^{18}O$ ratio analysis was only available at the bottom of spot 2 and spot 3 (here only $\delta^{34}S$). (b) Rayleigh plot for measured $SO_4^{2-}$ depletion at spot 2.**





**Figure 7: Concentration patterns and isotope ratios for CH₄ (a, b) and DIC (c), as well as average relative abundances of methanogens and methanotrophs (d).**



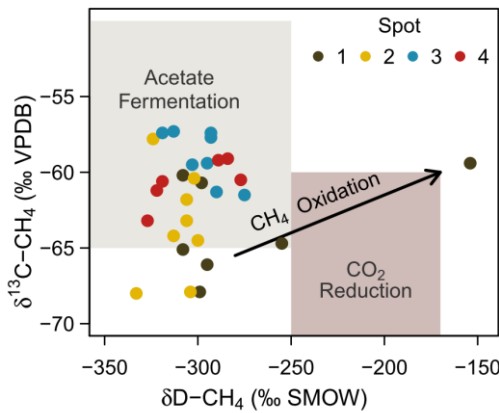

**Figure 8: Projection of the CH₄ stable isotope composition to differentiate dominating methanogenic pathways and methanotrophy.**
**Isotope thresholds to confine methanogenic pathways base on Whiticar et al. (1986). The concurrent increase in δ¹³C-CH₄ and δD-**
**CH₄ values at spot 1 suggests a downwards shift towards increasing CO₂ reduction or CH₄ oxidation rates at depth.**