# Peer review of "Sulfate deprivation triggers high methane production in a disturbed and rewetted coastal peatland"

_Biogeosciences, 2018_

## Referee Comment (RC1) · Anonymous Referee #1 · 18 Nov 2018

In the paper the effect of rewetting of agricultural peat field on methane formation has been studied by way of pore water and sediment chemistry, isotopic analysis as well as by studying the prevailing microbial community. The authors found evidence that rewetting by fresh water may increase methane emissions due to lack of sulfate and its reduction. In conclusion they suggested using marine water instead of fresh water in waterlogging.

1. The paper is in the scope of BG while it deals with interactions between biological and chemical processes in former cultivated field being subsequently wetland.

2. The authors have used versatile and state-of-art methods and the results are novel.

[Figure]

3. Substantial conclusions have been made. The supression of methanogenesis by sulfate reduction has been known for a long time. However, in this kind of practical context such substantial conclusions have not been made earlier.

4. The scientific methods are mainly clearly presented, with the exception of pH. Line 123: It was not stated that pH was measured even though the pH device was presented in the same sentence. In addition, the pH was presented in the principal analysis but not discussed. It is suggested that pH will be discussed.

5. The results are clearly presented and mainly in line with the text and the figures. However, there are discrepancies between the results in Table A1 and Figure 4.

Line 268: The authors write that "H2S concentrations were below detection limit ($\sim$1 $\mu$M, Fig. 4)". However, according to Table A1 there are higher concentrations (at 10 cm 3 $\mu$M, and at 40 cm 2 $\mu$m). In addition, the sulfate concentration is suggested to be reported with the same accuracy in Table A1 and Fig. 4. Otherwise the readers of the journal might get confused.

Line 274: The same comment as above regarding AVS, and similarly with the other spots.

6. The results support the interpretations and conclusions partly but I feel that the equivocal vertical methane concentrations do not clearly support the interpretations and conclusions.

For example, along the studied transect the sum of methane concentration till the depth of 40 cm is the highest in spot 2 and the lowest in spot 1. I feel that the topsoil of spot 1 might be aerated occasionally and therefore methanogen formation was the lowest there. The authors do not present any data for water levels in the spots although water saturation is crucial in determining whether the soil is aerobic or anaerobic.

In addition, in spot 2 the highest methane concentration is at the depth of 30 cm and there is still some sulfate left in pore water but not any methanogens. On the contrary,

at the depth of 30 cm and 20 cm methane concentrations are the lowest in the profile but in these layers there is not any sulfate left. This is contrary to the hypothesis and should be discussed in the text.

7. The description of experiments, the result table and calculations are sufficiently complete and precise to allow their reproduction by fellow scientists.

8. The authors refer to related work and clearly indicate their own new contribution. The title is clear and reflects the contents of the paper.

9. The abstract provides a concise and complete summary.

10. The overall presentation is well structured and clear.

11. The language is fluent and precise.

12. Mathematical formulae, symbols, abbreviations, and units are correctly defined and used.

13. The number and quality of references are appropriate.

14. The quality of supplementary material is appropriate, but the water table depths in the spots should be presented in a cross-section or in a table.

---

## Referee Comment (RC2) · Anonymous Referee #2 · 29 Dec 2018

The manuscript "Sulfate deprivation triggers high methane production in a disturbed and rewetted coastal peatland" by Koebsch et al. is very interesting and informative. The effect of dyking and freshwater rewetting on S and C transformation of a peatland has been studied using various analysis methods. The authors studied four spots with different solid S and sulfate concentrations and discussed the sulfate reduction and CH4 production and consumption among these sites. They conclude that a replenishment of marine water to dyked wetlands would reduce methane emissions.

Abstract:

Lines 30-32, the word of "suppression" is not accurate since results showed "high contents of labile iron minerals and dissolved ferrous iron at depth of spot 2, coincided with a high abundance of Thermodesulfovibrionaceae at concurrently minor occurrence of Deltaproteobacteria. (402-404)".

Lines 34-36: Is that useful to re-exposure of dyked wetlands to natural coastal dynamics since high amounts of sulfate did not interfere with high methane emissions on ecosystem scale?

Introduction

Lines 46-47: "dissimilatory sulfate reduction of dissolved organic matter (DSR)" changed to "dissimilatory sulfate reduction (DSR) of dissolved organic matter".

Line 63: Wrong left parenthesis. {Wen & Unger, 2018).

Line 75: Lack of "."

Material and Methods

Lines 105-107: Please check the parenthesis and should not be set in italic.

Line 159: Please define the abbreviation of DOC. It seems that the results of stable carbon isotope ratio of DOC was not presented in the result or discussion.

Line 160: Ertl and Spitzy (2004), not list in the Reference. Please check the manuscript completely.

Lines 168, 285, 308: Böttcher et al. (2007), not list in the Reference. Please check the manuscript completely.

Line 185: Please clarify DNA or RNA? Since only DNA has been extracted from samples, hence, I suggest to delete RNA.

Line 191: Please define the abbreviation of AVS. e.g. (acid volatile sulfur).

Lines 196-199: What's the meaning of "reactive iron"? How to detect the dissolved Fe and the valence of Fe (Fe(II) or (Fe(III)))? Line 200 indicated the extracted iron fraction

consists of iron(III) oxyhydroxides and iron(II) monosulfides, was the dissolved Fe(II) also existed (Heron et al., 1994. Speciation of Fe(II) and Fe(III) in contaminated aquifer sediments using chemical extraction techniques. Environmental Science & Technology, 28(9), 1698-705.)? The reference of Canfield, 1989 was not list, please check.

Line 200: Stookey (1970), not list in the Reference.

Line 223: "Sequence raw reads" changed to "Raw sequence reads".

Lines 229-230: Clarify the number of the sequence. 18.500 or 18,500? 12.500 or 12,500?

Results

Lines 240-241: Lack of ")". Please check the manuscript completely.

Lines 271-272: What's the detailed detection method of organic-bond S and where is the data of organic-bond S? Is it in the Figure 4 or TABLE A1? The note in the Figure 4 indicated that the solid residual S fraction is suggested to present primarily organic-bond S. However, the content (0.2 to 1.6%dwt) in spot 1 (lines 271-272) seems inconsistent with Figure 4b (far less than 1%dwt). What's the relationship between TABLE A1 and Figure 4 and 5ïij§

Lines 291, 322: Similar question to the line 271-272. Table A1 seems to address the organic-bond S content (0.2 to 1.6 %dwt). What's the difference between orgS in Table A1 and solid residual S in Figure 4b? Please clarify.

Lines 298-299 and 392-393: It seems inconsistent to Figure 4a. In the figure, the 1 mM sulfate concentration was appeared at 60cmbsf of spot 3, however, the relative abundance of Chloroflexi was missed at 60cmbsf of spot 3. Please clarify.

Lines 331-332: How to understand the role of DIC in methane production and consumption?

Line 340: Whiticar (1986), not list in the Reference. Please check.

Line 360: "acceptor" changed to "acceptors".

Discussion

Line 394: "dsrAB" should be italicized.

Line 439: Please define the abbreviation of AOM-SR.

Lines 447-448ïïjŽIn the Figure 7, the obvious increase of $\delta$13C-CH4 and $\delta$D-CH4 ratios were appeared at 50 cmsf of spot 1, however, the relative abundance of methanotrophy was missed at 50 cmsf, hence, how to understand this sentence.

Lines 452-453: Why high sulfate concentration did not interfere with methane production?

TABLE A1

Table A1: TS concentrations were inconsistent with dry weight ratios of TS. e.g. why the value of TS percentage is not 0 while TS concentration is 0. What is the meaning of orgS? Ionic valence should be added e.g. Cl-, Br-, Na+.

Reference

The format of the references should be carefully checked. Non-English language appeared e.g. lines 504-506, 523-524, 602-603. No journal e.g. lines 507-508, 566. Capitalize the first word of journal or not e.g. lines 548, 568, 573, 577. Other format errors in lines e.g. 523-524, 537-538, 546, 585, 624, 628.

FIGURES

Figure 2: Lines 645-646, please rewrite this sentence.

Figure 4: Please unify the formats with Figure 7. e.g. mmol l-1 in Figure 4 while mM in Figure 7. e.g. the name of y-coordinate...

---

## Author Comment (AC1) · 18 Jan 2019

Dear editor, dear reviewer,

We thank you for your thoughtful comments and the constructive suggestions which will be helpful to further improve the manuscript. Attached you can find our comments to those points that require a response as well as the suggested changes for the revised manuscript in bold print. We hope that we successfully addressed each point raised.

All the best Franziska Koebsch

[Figure]

Please also note the supplement to this comment:
https://www.biogeosciences-discuss.net/bg-2018-416/bg-2018-416-AC1-
supplement.pdf

―――――――――――――――――――――

[Figure]

**Supplement:**

Dear editor, dear reviewer,

We thank you for your thoughtful comments and the constructive suggestions which will be helpful to further improve the manuscript. Attached you can find our comments to those points that require a response as well as the suggested changes for the revised manuscript in bold print. We hope that we successfully addressed each point raised.

All the best

Franziska Koebsch

**Anonymous Referee #1**

In the paper the effect of rewetting of agricultural peat field on methane formation has been studied by way of pore water and sediment chemistry, isotopic analysis as well as by studying the prevailing microbial community. The authors found evidence that rewetting by fresh water may increase methane emissions due to lack of sulfate and its reduction. In conclusion they suggested using marine water instead of fresh water in waterlogging.

1. The paper is in the scope of BG while it deals with interactions between biological and chemical processes in former cultivated field being subsequently wetland.

2. The authors have used versatile and state-of-art methods and the results are novel.

3. Substantial conclusions have been made. The supression of methanogenesis by sulfate reduction has been known for a long time. However, in this kind of practical context such substantial conclusions have not been made earlier.

4. The scientific methods are mainly clearly presented, with the exception of pH. Line 123: It was not stated that pH was measured even though the pH device was presented in the same sentence. In addition, the pH was presented in the principal analysis but not discussed. It is suggested that pH will be discussed.

**Author's response**

- **You are correct, the pH measurements were not mentioned in the method section. Further, the patterns found in pH deserve to be discussed at least briefly.**

**Suggested change in the manuscript**

- **Information about pH measurements will be added in the methods section, further we will briefly discuss the observed patterns in pH with respect to its indicative value for peat degradation**

5. The results are clearly presented and mainly in line with the text and the figures. However, there are discrepancies between the results in Table A1 and Figure 4. Line 268: The authors write that "H2S concentrations were below detection limit ($\sim$1 µM, Fig. 4)". However, according to Table A1 there are higher concentrations (at 10 cm 3 µM, and at 40 cm 2 µm). In addition, the sulfate concentration is suggested to be reported with the same accuracy in Table A1 and Fig. 4. Otherwise the readers of

the journal might get confused. Line 274: The same comment as above regarding AVS, and similarly with the other spots.

**Author's response**

- **We agree, that the description of the H2S and AVS concentration were not very accurate and that the unit notations presented in Fig 4 (and Fig. 7a and 7c), and Table A1 require harmonization.**

**Suggested change in the manuscript**

- **We will change line 268 to "H2S concentrations hardly exceeded the detection limit (~1µM, Fig. 4)". Also inaccurate H2S and AVS quantity descriptions of other spots will be checked thoroughly.**
- **We will change the unit notation of Fig. 4a into 'mM dissolved S', the unit notation will then correspond with Table A1 and Fig. 7a and 7c**

6. The results support the interpretations and conclusions partly but I feel that the equivocal vertical methane concentrations do not clearly support the interpretations and conclusions. For example, along the studied transect the sum of methane concentration till the depth of 40 cm is the highest in spot 2 and the lowest in spot 1. I feel that the topsoil of spot 1 might be aerated occasionally and therefore methanogen formation was the lowest there. The authors do not present any data for water levels in the spots although water saturation is crucial in determining whether the soil is aerobic or anaerobic. In addition, in spot 2 the highest methane concentration is at the depth of 30 cm and there is still some sulfate left in pore water but not any methanogens. On the contrary, at the depth of 30 cm and 20 cm methane concentrations are the lowest in the profile but in these layers there is not any sulfate left. This is contrary to the hypothesis and should be discussed in the text.

**Author's response**

- **We agree that patterns in methane concentrations are equivocal. In general, methane concentration pattern should be Interpreted with care as methane is a highly volatile gas. Especially at high methane production rates, the indicative value of methane concentration profiles can be easily impaired by erratic ebullitive release. Hence, the observed methane concentration patterns are likely to present a snapshot resulting from the combination of methane production and erratic methane loss but may not be well suited to represent overall patterns in methane production. Therefore, we are very careful concerning the indicative value of single methane concentration data points. Still, we decided to show the methane concentration profiles for the sake of completeness.**
- **Since methanogenesis exerts a strong fractionating effect on $CO_2$ and DIC is less volatile than methane, we use the $\delta^{13}C$ values of DIC as indicator for methane production. In concert, with the isotopic composition of methane and the microbial structure, these can provide a comprehensive picture on methane cycling in our study site.**
- **In the current manuscript, we have explicitly qualified the indicative value of methane concentrations and explained why we focus on the isotope composition of methane and DIC instead in line 116f: "Measured pore water $CH_4$ concentrations were up to 0.7 mM with equivocal vertical patterns across spots (Fig. 7a), reflecting the methane-specific spatial variability that evolves from small-scale heterogeneity in production and consumption processes and from ebullitive release events (Chanton et al., 1989; Whalen, 2005). The isotope composition of $CH_4$ (Fig. 7b) and DIC (Fig. 7c) provided a clearer (and probably more robust) indication for patterns of methanogenesis and methanotrophy".**

- **Referee 1 is right in his suggestion that spot 1 is located on slightly higher grounds. Indeed, since the rewetting of the wetland, all spots have been flooded throughout the year, so contemporary water levels should not restrict methane production. In the current manuscript, we have mentioned the hydrological state in line 116f: "At the time of sampling, water depth above peat surface spanned from 15 to 25 cm, which presented the lowest range within the seasonal water level fluctuation". In fact, lower water levels in the past have, in combination with groundwater flow from the nearby forest catchment, certainly affected peat formation and soil geochemistry at spot 1.**

**Suggested change in the manuscript**

- **We will add measured water tables in table A1 and remind the reader about permanently inundated conditions in the results and discussion section. Further, we will discuss lower water levels in the past, in combination with groundwater flow from the nearby forest catchment, as possible reasons for the specific geochemistry and microbial community at spot 1.**

7. The description of experiments, the result table and calculations are sufficiently complete and precise to allow their reproduction by fellow scientists.

8. The authors refer to related work and clearly indicate their own new contribution. The title is clear and reflects the contents of the paper.

9. The abstract provides a concise and complete summary.

10. The overall presentation is well structured and clear.

11. The language is fluent and precise.

12. Mathematical formulae, symbols, abbreviations, and units are correctly defined and used.

13. The number and quality of references are appropriate.

14. The quality of supplementary material is appropriate, but the water table depths in the spots should be presented in a cross-section or in a table.

**Suggested change in the manuscript**

- **We will add measured water tables in table A1.**

---

## Author Comment (AC2) · 18 Jan 2019

Dear editor, dear reviewer,

We thank you for your thoughtful comments and the constructive suggestions which will be helpful to further improve the manuscript. Attached you can find our comments to those points that require a response as well as the suggested changes for the revised manuscript in bold print. We hope that we successfully addressed each point raised. References used in the author comments are listed at the end of the document.

All the best Franziska Koebsch

Please also note the supplement to this comment:
https://www.biogeosciences-discuss.net/bg-2018-416/bg-2018-416-AC2-supplement.pdf

**Supplement:**

Dear editor, dear reviewer,

We thank you for your thoughtful comments and the constructive suggestions which will be helpful to further improve the manuscript. Attached you can find our comments to those points that require a response as well as the suggested changes for the revised manuscript in bold print. We hope that we successfully addressed each point raised. References used in the author comments are listed at the end of the document.

All the best

Franziska Koebsch

**Anonymous Referee #2**

The manuscript "Sulfate deprivation triggers high methane production in a disturbed and rewetted coastal peatland" by Koebsch et al. is very interesting and informative. The effect of dyking and freshwater rewetting on S and C transformation of a peatland has been studied using various analysis methods. The authors studied four spots with different solid S and sulfate concentrations and discussed the sulfate reduction and CH4 production and consumption among these sites. They conclude that a replenishment of marine water to dyked wetlands would reduce methane emissions.

Abstract:

Lines 30-32, the word of "suppression" is not accurate since results showed "high contents of labile iron minerals and dissolved ferrous iron at depth of spot 2, coincided with a high abundance of Thermodesulfovibrionaceae at concurrently minor occurrence of Deltaproteobacteria. (402-404)".

Lines 34-36: Is that useful to re-exposure of dyked wetlands to natural coastal dynamics since high amounts of sulfate did not interfere with high methane emissions on ecosystem scale?

**Author's response**

- **Indeed, we think that dissimilatory sulfate reduction at spot 2 is inhibited because we suggest *Thermodesulfovibrionaceae* present at this spot to utilize mainly ferric iron as electron acceptor (Fortney et al., 2016). This is indicated by the long-term persistence of the brackish sulfate reservoir at concurrently high contents of labile iron minerals and dissolved ferrous iron. This interpretation is stated in detail in line 394f. of the manuscript**
- **We admit that the used formulations in line 34-36 may be confusing for the reader.**

**Suggested change in the manuscript**

- **We will replace 'suppression' in line 30-32 with 'inhibition'**
- **We will rephrase the sentences in line 34-36 accordingly. In particular, we will emphasize that the local sulfate remnants did not interfere with high methane emissions on ecosystem scale as sulfate was spatially confined to depths below the methane production zone. In contrast, the re-exposure to coastal dynamics would replenish the brackish sulfate reservoir within the entire wetland and is therefore likely to lower methane emissions on ecosystem scale.**

Introduction

Lines 46-47: "dissimilatory sulfate reduction of dissolved organic matter (DSR)" changed to "dissimilatory sulfate reduction (DSR) of dissolved organic matter".

Line 63: Wrong left parenthesis. {Wen & Unger, 2018).

Line 75: Lack of "."

**Author's response**

- **Text mistakes will be corrected accordingly**

Material and Methods

Lines 105-107: Please check the parenthesis and should not be set in italic.

Line 159: Please define the abbreviation of DOC. It seems that the results of stable carbon isotope ratio of DOC was not presented in the result or discussion.

Line 160: Ertl and Spitzy (2004), not list in the Reference. Please check the manuscript completely.

Lines 168, 285, 308: Böttcher et al. (2007), not list in the Reference. Please check the manuscript completely.

Line 185: Please clarify DNA or RNA? Since only DNA has been extracted from samples, hence, I suggest to delete RNA.

Line 191: Please define the abbreviation of AVS. e.g. (acid volatile sulfur).

Lines 196-199: What's the meaning of "reactive iron"? How to detect the dissolved Fe and the valence of Fe (Fe(II) or (Fe(III))? Line 200 indicated the extracted iron fraction consists of iron(III) oxyhydroxides and iron(II) monosulfides, was the dissolved Fe(II) also existed (Heron et al., 1994. Speciation of Fe(II) and Fe(III) in contaminated aquifer sediments using chemical extraction techniques. Environmental Science & Technology, 28(9), 1698-705.)? The reference of Canfield, 1989 was not list, please check.

Line 200: Stookey (1970), not list in the Reference.

Line 223: "Sequence raw reads" changed to "Raw sequence reads".

Lines 229-230: Clarify the number of the sequence. 18.500 or 18,500? 12.500 or 12,500?

**Author's response**

- **Reactive iron is considered here as the sum of those iron fractions that still may react with dissolved sulfide. The applied analytical extraction scheme extracts the sum of remaining iron(III) oxyhydroxides and acid volatile sulfide (AVS, essentially FeS) as well as a very minor contribution from dissolved Fe2+ in the pore water (Canfield, 1989). No further quantification of different di- and trivalent iron was carried out.**

**Suggested change in the manuscript**

- **The meaning of 'reactive iron' will be specified in the method section and the reference Canfield 1989 will be added in the reference list**
- **Text mistakes will be corrected accordingly, references will be checked, abbreviations will be defined**

Results

Lines 240-241: Lack of ")". Please check the manuscript completely.

Lines 271-272: What's the detailed detection method of organic-bond S and where is the data of organic-bond S? Is it in the Figure 4 or TABLE A1? The note in the Figure 4 indicated that the solid residual S fraction is suggested to present primarily organic-bond S. However, the content (0.2 to 1.6%dwt) in spot 1 (lines 271-272) seems inconsistent with Figure 4b (far less than 1%dwt). What's the relationship between TABLE A1 and Figure 4 and 5ïij§

Lines 291, 322: Similar question to the line 271-272. Table A1 seems to address the organic-bond S content (0.2 to 1.6 %dwt). What's the difference between orgS in Table A1 and solid residual S in Figure 4b? Please clarify.

Lines 298-299 and 392-393: It seems inconsistent to Figure 4a. In the figure, the 1 mM sulfate concentration was appeared at 60cmbsf of spot 3, however, the relative abundance of Chloroflexi was missed at 60cmbsf of spot 3. Please clarify.

Lines 331-332: How to understand the role of DIC in methane production and consumption?

Line 340: Whiticar (1986), not list in the Reference. Please check.

**Author's response**

- **In our study we assume that the solid residual S fraction corresponds primarily to organic-bond S. Unfortunately, we have made a mistake in the residual S fraction values in table A1. The mistake will be corrected. However, this will not affect the main patterns and results of our work.**
- **We are sorry that the depth indications for spot 3 in Fig. 4d are shifted. The text is correct as the 40% relative abundance of Chloroflexi of the class Dehalococcoidetes represented the dominating bacterial group at the 1 mM sulfate concentration at 60 cm depth of spot 3.**
- **The formation of methane is a highly fractionating process and results in $CO_2$ that is considerably enriched in $^{13}C$ compared to the starting organic material (~-27‰ in this study) (Whiticar et al. 1986). The isotopic composition of DIC-C can therefore be used as indicator for methane formation in relation to non-fractionating pathways (organic matter fermentation, sulfate reduction, e. g. Corbett et al. 2013). Dissolved methane is highly volatile and especially at high methane production rates, the indicative value of methane concentration profiles can be easily impaired by erratic ebullitive release. Therefore, in this study we decided to use the $\delta^{13}C$ values of DIC as indicator for methane production. In concert, with the microbial structure, these can provide a comprehensive picture on methane cycling in our study site.**

**Suggested change in the manuscript**

- **the residual S fraction values in table A1 will be corrected and then correspond to the organic bond S values in Fig. 4. The text will be checked thoroughly for mistakes.**
- **The depth indications for Fig. 4d spot 3 will be corrected.**
- **Text mistakes will be corrected accordingly, references will be checked.**
- **We will add a note on the indicative value of $\delta^{13}C$-DIC for methanogenesis in the text.**
- **Text mistakes will be corrected accordingly, references will be checked**

Discussion

Line 394: "dsrAB" should be italicized.

Line 439: Please define the abbreviation of AOM-SR.

Lines447-448ïïjŽ In the Figure 7, the obvious increase of δ13C-CH4 and δD-CH4ratios were appeared at 50 cmsf of spot 1, however, the relative abundance of methanotrophy was missed at 50 cmsf, hence, how to understand this sentence.

Lines 452-453: Why high sulfate concentration did not interfere with methane production?

**Author's response**

- **Likewise with Fig 4d, depth indications for Fig. 7d are accidentally shifted. The 48% abundance of ANME-2d ranges down to a depth of 50 cmbsf. Accordingly, the observed patterns in δD-CH$_4$ and δ$^{13}$C-CH$_4$ are consistent with the microbial community structure.**

- **The remnants of the brackish sulfate reservoir were spatially separated from the zones of methane production: As a result of intense dissimilatory sulfate reduction, residual sulfate persisted only at depth of spot 2 (and in lower amounts at depth of spot 3). Methane production in wetlands is usually confined to the upper peat horizons, and also in our study methane was mainly produced in the upper 30-40 cm. Therefore, we assume that the residual sulfate does not significantly affect methane production. This is stated in line 452-459:** *As intense methane production was confined to the upper pore water layers in the entire peatland, it did not interfere with high sulfate concentrations locally preserved as legacy of former brackish impact in the bottom.*

**Suggested change in the manuscript**

- **Text mistakes will be corrected accordingly, abbreviations will be defined**
- **The depth indications for Fig. 7d will be corrected.**

Table A1: TS concentrations were inconsistent with dry weight ratios of TS. e.g. why the value of TS percentage is not 0 while TS concentration is 0. What is the meaning of orgS? Ionic valence should be added e.g. Cl-, Br-, Na+.

**Author's response**

- **TS in mmol corresponds to the total dissolved S fraction whilst TS in dry weight ratios corresponds to the solid dissolved S fraction. Anyway, we admit that it would be better to use different abbreviations to avoid confusion.**
- **Organic S is the residual non-specified solid S fraction.**

**Suggested change in the manuscript**

- **We will implement different abbreviations for solid and dissolved total S**
- **As stated above, we will correct the values of orgS so that they are consistent with the quantities represented in Fig 4**
- **Ionic valences will be added**

Reference

The format of the references should be carefully checked. Non-English language appeared e.g. lines 504-506, 523-524, 602-603. No journal e.g. lines 507-508, 566. Capitalize the first word of journal or not e.g. lines 548, 568, 573, 577. Other format errors in lines e.g. 523-524, 537-538, 546, 585, 624, 628.

**Suggested change in the manuscript**

- **The reference list will be carefully checked**

Figures

Figure 2: Lines 645-646, please rewrite this sentence.

Figure 4: Please unify the formats with Figure 7. e.g. mmol l-1 in Figure 4 while mM in Figure 7. e.g. the name of y-coordinate.

**Suggested change in the manuscript**

- **Figure captions and axis labels will be corrected accordingly**

**Author's response reference list**

Corbett, J. E., Tfaily, M. M., Burdige, D. J., Cooper, W. T., Glaser, P. H., & Chanton, J. P. (2013). Partitioning pathways of CO2 production in peatlands with stable carbon isotopes. *Biogeochemistry*, *114*(1-3), 327-340.

Fortney, N. W., He, S., Converse, B. J., Beard, B. L., Johnson, C. M., Boyd, E. S., & Roden, E. E. (2016). Microbial Fe (III) oxide reduction potential in Chocolate Pots hot spring, Yellowstone National Park. *Geobiology*, *14*(3), 255-275.

Whiticar, M. J., Faber, E., & Schoell, M. (1986). Biogenic methane formation in marine and freshwater environments: CO2 reduction vs. acetate fermentation—isotope evidence. *Geochimica et Cosmochimica Acta*, *50*(5), 693-709.

---

## Author Response (AR1)

Dear editor, dear reviewer,

We thank you for your thoughtful comments and the constructive suggestions which were helpful to further improve the manuscript. Attached you can find our comments to those points that require a response, the suggested changes for the revised manuscript in bold print, and a marked-up version of the manuscript. We hope that we successfully addressed each point raised.

All the best
Franziska Koebsch

Anonymous Referee #1

In the paper the effect of rewetting of agricultural peat field on methane formation has been studied by way of pore water and sediment chemistry, isotopic analysis as well as by studying the prevailing microbial community. The authors found evidence that rewetting by fresh water may increase methane emissions due to lack of sulfate and its reduction. In conclusion they suggested using marine water instead of fresh water in waterlogging.

1. The paper is in the scope of BG while it deals with interactions between biological and chemical processes in former cultivated field being subsequently wetland.

2. The authors have used versatile and state-of-art methods and the results are novel.

3. Substantial conclusions have been made. The supression of methanogenesis by sulfate reduction has been known for a long time. However, in this kind of practical context such substantial conclusions have not been made earlier.

4. The scientific methods are mainly clearly presented, with the exception of pH. Line 123: It was not stated that pH was measured even though the pH device was presented in the same sentence. In addition, the pH was presented in the principal analysis but not discussed. It is suggested that pH will be discussed.

**Author's response**
- **You are correct, the pH measurements were not mentioned in the method section. Further, the patterns found in pH deserve to be discussed at least briefly.**

**Changes in the manuscript**
- **pH measurements were mentioned in the methods section, pH values were added in table A1 and the observed patterns in pH are integrated in the description of overall pore water geochemical patterns.**

5. The results are clearly presented and mainly in line with the text and the figures. However, there are discrepancies between the results in Table A1 and Figure 4. Line 268: The authors write that "H2S concentrations were below detection limit ($\sim$1 µM, Fig. 4)". However, according to Table A1 there are higher concentrations (at 10 cm 3 µM, and at 40 cm 2 µm). In addition, the sulfate concentration is suggested to be reported with the same accuracy in Table A1 and Fig. 4. Otherwise the readers of the journal might get confused. Line 274: The same comment as above regarding AVS, and similarly with the other spots.

**Author's response**
- **We agree, that the description of the H2S and AVS concentration were not very accurate and that the unit notations presented in Fig 4 (and Fig. 7a and 7c), and Table A1 require harmonization.**

**Changes in the manuscript**
- **Line 268 (now line 279) was changed to "H2S concentrations hardly exceeded the detection limit ($\sim$1µM, Fig. 4)". inaccurate H2S and AVS quantity descriptions of other spots were checked thoroughly.**

- **We changed the unit notation of Fig. 4a into 'mM dissolved S', the unit notation now corresponds with Table A1 and Fig. 7a and 7c.**

6. The results support the interpretations and conclusions partly but I feel that the equivocal vertical methane concentrations do not clearly support the interpretations and conclusions. For example, along the studied transect the sum of methane concentration till the depth of 40 cm is the highest in spot 2 and the lowest in spot 1. I feel that the topsoil of spot 1 might be aerated occasionally and therefore methanogen formation was the lowest there. The authors do not present any data for water levels in the spots although water saturation is crucial in determining whether the soil is aerobic or anaerobic. In addition, in spot 2 the highest methane concentration is at the depth of 30 cm and there is still some sulfate left in pore water but not any methanogens. On the contrary, at the depth of 30 cm and 20 cm methane concentrations are the lowest in the profile but in these layers there is not any sulfate left. This is contrary to the hypothesis and should be discussed in the text.

**Author's response**

- **We agree that patterns in methane concentrations are equivocal. In general, methane concentration pattern should be Interpreted with care as methane is a highly volatile gas. Especially at high methane production rates, the indicative value of methane concentration profiles can be easily impaired by erratic ebullitive release. Hence, the observed methane concentration patterns are likely to present a snapshot resulting from the combination of methane production and erratic methane loss but may not be well suited to represent overall patterns in methane production. Therefore, we are very careful concerning the indicative value of single methane concentration data points. Still, we decided to show the methane concentration profiles for the sake of completeness.**
- **Since methanogenesis exerts a strong fractionating effect on $CO_2$ and DIC is less volatile than methane, we use the $\delta^{13}C$ values of DIC as indicator for methane production. In concert, with the isotopic composition of methane and the microbial structure, these can provide a comprehensive picture on methane cycling in our study site.**
- **In the current manuscript, we have explicitly qualified the indicative value of methane concentrations and explained why we focus on the isotope composition of methane and DIC instead in line 342f.: "Measured pore water $CH_4$ concentrations were up to 643 µM with equivocal vertical patterns across spots (Fig. 7a (Fig. 7a), reflecting the methane-specific spatial variability that evolves from small-scale heterogeneity in production and consumption processes and from ebullitive release events (Chanton *et al.*, 1989, Whalen, 2005). The isotope composition of $CH_4$ (Fig. 7b) and DIC (Fig. 7c) provided a clearer (and probably more robust) indication for patterns of methanogenesis and methanotrophy".**

- Referee 1 is right in his suggestion that spot 1 is located on slightly higher grounds than spots 2 and 3. Indeed, since the rewetting of the wetland, all spots have been flooded throughout the year, so contemporary water levels should not restrict methane production. In the current manuscript, we have mentioned the hydrological state in line 118f: "At the time of sampling, water depth above peat surface spanned from 9 to 19 cm, which presented the lowest range within the seasonal water level fluctuation". In fact, lower water levels in the past have, in combination with groundwater flow from the nearby forest catchment, certainly affected peat formation and soil geochemistry at spot 1.

**Changes in the manuscript**
- **We added measured water tables in table A1 and reminded the reader about permanently inundated conditions in the discussion section (line 391). Further, we discussed lower water levels in the past, in combination with mineral inflow from the nearby forest catchment, as possible reasons for the specific geochemistry and microbial community at spot 1 (line 468f.).**

7. The description of experiments, the result table and calculations are sufficiently complete and precise to allow their reproduction by fellow scientists.
8. The authors refer to related work and clearly indicate their own new contribution. The title is clear and reflects the contents of the paper.
9. The abstract provides a concise and complete summary.
10. The overall presentation is well structured and clear.
11. The language is fluent and precise.
12. Mathematical formulae, symbols, abbreviations, and units are correctly defined and used.
13. The number and quality of references are appropriate.
14. The quality of supplementary material is appropriate, but the water table depths in the spots should be presented in a cross-section or in a table.

**Changes in the manuscript**
- **We added water tables in table A1.**

Anonymous Referee #2

The manuscript "Sulfate deprivation triggers high methane production in a disturbed and rewetted coastal peatland" by Koebsch et al. is very interesting and informative. The effect of dyking and freshwater rewetting on S and C transformation of a peatland has been studied using various analysis methods. The authors studied four spots with different solid S and sulfate concentrations and discussed the sulfate reduction and CH4 production and consumption among these sites. They conclude that a replenishment of marine water to dyked wetlands would reduce methane emissions.

Abstract:

Lines 30-32, the word of "suppression" is not accurate since results showed "high contents of labile iron minerals and dissolved ferrous iron at depth of spot 2, coincided with a high abundance of Thermodesulfovibrionaceae at concurrently minor occurrence of Deltaproteobacteria. (402-404)".

Lines 34-36: Is that useful to re-exposure of dyked wetlands to natural coastal dynamics since high amounts of sulfate did not interfere with high methane emissions on ecosystem scale?

**Author's response**

- **Indeed, we think that dissimilatory sulfate reduction at spot 2 is inhibited because we suggest *Thermodesulfovibrionaceae* present at this spot to utilize mainly ferric iron as electron acceptor (Fortney *et al.*, 2016). This is indicated by the long-term persistence of the brackish sulfate reservoir at concurrently high contents of labile iron minerals and dissolved ferrous iron. This interpretation is stated in detail in line 425 (formerly 394) of the manuscript.**
- **We admit that the used formulations in former lines 34-36 might have been confusing for the reader.**

**Changes in the manuscript**

- **'suppression' in line 29 (formerly lines 30-32) is replaced with 'inhibition'.**
- **we added an explanation in line 30f.: *However, as the occurrence of sulfate was confined to the peat layers below 30-40 cm, it did not interfere with high methane emissions on ecosystem scale*.**

Introduction

Lines 46-47: "dissimilatory sulfate reduction of dissolved organic matter (DSR)" changed to "dissimilatory sulfate reduction (DSR) of dissolved organic matter".

Line 63: Wrong left parenthesis. {Wen & Unger, 2018).

Line 75: Lack of "."

**Author's response**

**Text mistakes were corrected accordingly.**

Material and Methods

Lines 105-107: Please check the parenthesis and should not be set in italic.

Line 159: Please define the abbreviation of DOC. It seems that the results of stable carbon isotope ratio of DOC was not presented in the result or discussion.

Line 160: Ertl and Spitzy (2004), not list in the Reference. Please check the manuscript completely.

Lines 168, 285, 308: Böttcher et al. (2007), not list in the Reference. Please check the manuscript completely.

Line 185: Please clarify DNA or RNA? Since only DNA has been extracted from samples, hence, I suggest to delete RNA.

Line 191: Please define the abbreviation of AVS. e.g. (acid volatile sulfur).

Lines 196-199: What's the meaning of "reactive iron"? How to detect the dissolved Fe and the valence of Fe (Fe(II) or (Fe(III))?

Line 200 indicated the extracted iron fraction consists of iron(III) oxyhydroxides and iron(II) monosulfides, was the dissolved Fe(II) also existed (Heronet al., 1994. Speciation of Fe(II) and Fe(III) in contaminated aquifer sediments using chemical extraction techniques. Environmental Science & Technology, 28(9), 1698-705.)? The reference of Canfield, 1989 was not list, please check.

Line 200: Stookey (1970), not list in the Reference.

Line 223: "Sequence raw reads" changed to "Raw sequence reads".

Lines 229-230: Clarify the number of the sequence. 18.500 or 18,500? 12.500 or 12,500?

**Author's response**

- **Reactive iron is considered here as the sum of those iron fractions that still may react with dissolved sulfide. The applied analytical extraction scheme extracts the sum of remaining iron(III) oxyhydroxides and acid volatile sulfide (AVS, essentially FeS) as well as a very minor contribution from dissolved Fe2+ in the pore water (Canfield, 1989). No further quantification of different di- and trivalent iron was carried out.**

**Changes in the manuscript**

- **The meaning of 'reactive iron' was specified in the method section and the reference Canfield 1989 was added in the reference list.**

- **Text mistakes were corrected accordingly, references were checked, abbreviations were defined.**

Results

Lines 240-241: Lack of ")". Please check the manuscript completely.

Lines 271-272: What's the detailed detection method of organic-bond S and where is the data of organic-bond S? Is it in the Figure 4 or TABLE A1? The note in the Figure 4 indicated that the solid residual S fraction is suggested to present primarily organic-bond S. However, the content (0.2 to 1.6%dwt) in spot 1 (lines 271-272) seems inconsistent with Figure 4b (far less than 1%dwt). What's the relationship between TABLE A1 and Figure 4 and 5ïïj§

Lines 291, 322: Similar question to the line 271-272. Table A1 seems to address the organic-bond S content (0.2 to 1.6 %dwt). What's the difference between orgS in Table A1 and solid residual S in Figure 4b? Please clarify.

Lines 298-299 and 392-393: It seems inconsistent to Figure 4a. In the figure, the 1 mM sulfate concentration was appeared at 60cmbsf of spot 3, however, the relative abundance of Chloroflexi was missed at 60cmbsf of spot 3. Please clarify.

Lines 331-332: How to understand the role of DIC in methane production and consumption?

Line 340: Whiticar (1986), not list in the Reference. Please check.

**Author's response**

- **In our study we assume that the solid residual S fraction corresponds primarily to organic-bond S. Unfortunately, we have made a mistake in the residual S fraction values in table A1. The mistake will be corrected. However, this will not affect the main patterns and results of our work.**

- **We are sorry that the depth indications for spot 3 in Fig. 4d are shifted. The text is correct as the 40% relative abundance of Chloroflexi of the class Dehalococcoidetes represented the dominating bacterial group at the 1 mM sulfate concentration at 60 cm depth of spot 3.**

- **The formation of methane is a highly fractionating process and results in $CO_2$ that is considerably enriched in $^{13}C$ compared to the starting organic material (~-27‰ in this study) (Whiticar et al. 1986). The isotopic composition of DIC-C can therefore be used as indicator for methane formation in relation to non-fractionating pathways (organic matter fermentation, sulfate reduction, e. g. Corbett et al. 2013). Dissolved methane is highly volatile and especially at high methane production rates, the indicative value of methane concentration profiles can be easily impaired by erratic ebullitive release. Therefore, in this study we decided to use the $\delta^{13}C$ values of DIC as indicator for methane production. In concert, with the microbial structure, these can provide a comprehensive picture on methane cycling in our study site.**

**Changes in the manuscript**

- **the residual S fraction values in table A1 were corrected and then correspond to the organic bond S values in Fig. 4. The text was checked thoroughly for mistakes.**

- **The depth indications for Fig. 4d spot 3 were corrected.**

- **Text mistakes were corrected accordingly, references were checked.**

- **We added a note on the indicative value of $\delta^{13}C$-DIC for methanogenesis in the text (lines 343f.).**

Discussion

Line 394: "dsrAB" should be italicized.

Line 439: Please define the abbreviation of AOM-SR.

Lines447-448ïjŽ In the Figure 7, the obvious increase of δ13C-CH4 and δD-CH4ratios were appeared at 50 cmsf of spot 1, however, the relative abundance of methanotrophy was missed at 50 cmsf, hence, how to understand this sentence.

Lines 452-453: Why high sulfate concentration did not interfere with methane production?

**Author's response**

- **Likewise with Fig 4d, depth indications for Fig. 7d are accidentally shifted. The 48% abundance of ANME-2d ranges down to a depth of 50 cmbsf. Accordingly, the observed patterns in δD-CH₄ and δ¹³C-CH₄ are consistent with the microbial community structure.**
- **The remnants of the brackish sulfate reservoir were spatially separated from the zones of methane production: As a result of intense dissimilatory sulfate reduction, residual sulfate persisted only at depth of spot 2 (and in lower amounts at depth of spot 3). Methane production in wetlands is usually confined to the upper peat horizons, and also in our study methane was mainly produced in the upper 30-40 cm. Therefore, we assume that the residual sulfate does not significantly affect methane production. This is stated in line 473f.: *As intense methane production was confined to the upper pore water layers in the entire peatland, it did not interfere with high sulfate concentrations locally preserved as legacy of former brackish impact in the bottom.***

**Changes in the manuscript**

- **Text mistakes were corrected accordingly, abbreviations were defined.**
- **The depth indications for Fig. 7d were corrected.**

Table A1: TS concentrations were inconsistent with dry weight ratios of TS. e.g. why the value of TS percentage is not 0 while TS concentration is 0. What is the meaning of orgS? Ionic valence should be added e.g. Cl-, Br-, Na+.

**Author's response**

- **TS in mmol corresponds to the total dissolved S fraction whilst TS in dry weight ratios corresponds to the solid dissolved S fraction. Anyway, we admit that it would be better to use different abbreviations to avoid confusion.**
- **Organic S is the residual non-specified solid S fraction.**

**Changes in the manuscript**

- **We implemented different abbreviations for solid and dissolved total S.**
- **As stated above, we corrected the values of orgS so that they are consistent with the quantities represented in Fig 4.**
- **Ionic valences were added .**

Reference

The format of the references should be carefully checked. Non-English language appeared e.g. lines 504-506, 523-524, 602-603. No journal e.g. lines 507-508, 566. Capitalize the first word of journal or not e.g. lines 548, 568, 573, 577. Other format errors in lines e.g. 523-524, 537-538, 546, 585, 624, 628.

**Changes in the manuscript**

- **The reference list was carefully checked.**

Figures

Figure 2: Lines 645-646, please rewrite this sentence.

Figure 4: Please unify the formats with Figure 7. e.g. mmol l-1 in Figure 4 while mM in Figure 7. e.g. the name of y-coordinate.

**Changes in the manuscript**

- **Figure captions and axis labels were corrected accordingly.**

**Author's response reference list**

**Corbett, J. E., Tfaily, M. M., Burdige, D. J., Cooper, W. T., Glaser, P. H., & Chanton, J. P. (2013). Partitioning pathways of CO2 production in peatlands with stable carbon isotopes. *Biogeochemistry*, *114*(1-3), 327-340.**

**Fortney, N. W., He, S., Converse, B. J., Beard, B. L., Johnson, C. M., Boyd, E. S., & Roden, E. 
[revised manuscript text omitted]

Böttcher ME, Lepland A (2000) Biogeochemistry of sulfur in a sediment core from the west-central Baltic Sea: evidence from stable isotopes and pyrite textures. Journal of Marine Systems, 25, 299-312.

Brand WA, Coplen TB (2012) Stable isotope deltas: tiny, yet robust signatures in nature. Isotopes in environmental and health studies, 48, 393-409.

Brown K, Macqueen J (1985) Sulphate uptake from surface water by peat. Soil Biology and Biochemistry, 17, 411-420.

Butler IB, Böttcher ME, Rickard D, Oldroyd A (2004) Sulfur isotope partitioning during experimental formation of pyrite via the polysulfide and hydrogen sulfide pathways: implications for the interpretation of sedimentary and hydrothermal pyrite isotope records. Earth and Planetary Science Letters, 228, 495-509.

Canfield DE (1989) Reactive iron in marine sediments. Geochimica et Cosmochimica Acta, 53, 619-632.

Canfield DE (2001) Isotope fractionation by natural populations of sulfate-reducing bacteria. Geochimica et Cosmochimica Acta, 65, 1117-1124.

Caporaso JG, Kuczynski J, Stombaugh J et al. (2010) QIIME allows analysis of high-throughput community sequencing data. Nature methods, 7, 335.

Chanton JP, Martens CS, Kelley CA (1989) Gas transport from methane-saturated, tidal freshwater and wetland sediments. Limnology and Oceanography, 34, 807-819.

Cline JD (1969) Spectrophotometric determination of hydrogen sulfide in natural waters 1. Limnology and Oceanography, 14, 454-458.

Dahms P (1991) Studie Wasserregulierung Hütelmoor. Universität Rostock, Fachbereich Landeskultur und Umweltschutz, Fachgebiet Kulturtechnik.

Deverel SJ, Ingrum T, Leighton D (2016) Present-day oxidative subsidence of organic soils and mitigation in the Sacramento-San Joaquin Delta, California, USA. Hydrogeology journal, 24, 569-586.

Deverel SJ, Rojstaczer S (1996) Subsidence of agricultural lands in the Sacramento-San Joaquin Delta, California: Role of aqueous and gaseous carbon fluxes. Water Resources Research, 32, 2359-2367.

Erkens G, Van Der Meulen MJ, Middelkoop H (2016) Double trouble: subsidence and CO 2 respiration due to 1,000 years of Dutch coastal peatlands cultivation. Hydrogeology journal, 24, 551-568.

Ettwig KF, Zhu B, Speth D, Keltjens JT, Jetten MS, Kartal B (2016) Archaea catalyze iron-dependent anaerobic oxidation of methane. Proceedings of the National Academy of Sciences, 113, 12792-12796.

Evans PN, Parks DH, Chadwick GL, Robbins SJ, Orphan VJ, Golding SD, Tyson GW (2015) Methane metabolism in the archaeal phylum Bathyarchaeota revealed by genome-centric metagenomics. Science, 350, 434-438.

Feistel R, Weinreben S, Wolf H et al. (2010) Density and absolute salinity of the Baltic Sea 2006–2009. Ocean Science, 6, 3-24.

Fortney N, He S, Converse B, Beard B, Johnson C, Boyd ES, Roden E (2016) Microbial F e (III) oxide reduction potential in C hocolate P ots hot spring, Y ellowstone N ational P ark. Geobiology, 14, 255-275.

[revised manuscript text omitted]